# Fate mapping of hematopoietic stem cells reveals two pathways of native thrombopoiesis

Mina N. F. Morcos [1,7], Congxin Li[2,6,7], Clara M. Munz[1,7], Alessandro Greco [2,3], Nicole Dressel[1], Susanne Reinhardt [4], Katrin Sameith [4], Andreas Dahl [4], Nils B. Becker[2], Axel Roers [5], Thomas Höfer [2,3✉] & Alexander Gerbaulet [1✉]

Hematopoietic stem cells (HSCs) produce highly diverse cell lineages. Here, we chart native lineage pathways emanating from HSCs and define their physiological regulation by computationally integrating experimental approaches for fate mapping, mitotic tracking, and single-cell RNA sequencing. We find that lineages begin to split when cells leave the tip HSC population, marked by high Sca-1 and CD201 expression. Downstream, HSCs either retain high Sca-1 expression and the ability to generate lymphocytes, or irreversibly reduce Sca-1 level and enter into erythro-myelopoiesis or thrombopoiesis. Thrombopoiesis is the sum of two pathways that make comparable contributions in steady state, a long route via multipotent progenitors and CD48$^{hi}$ megakaryocyte progenitors (MkPs), and a short route from HSCs to developmentally distinct CD48$^{-/lo}$ MkPs. Enhanced thrombopoietin signaling differentially accelerates the short pathway, enabling a rapid response to increasing demand. In sum, we provide a blueprint for mapping physiological differentiation fluxes from HSCs and decipher two functionally distinct pathways of native thrombopoiesis.

[1] Institute for Immunology, Faculty of Medicine, TU Dresden, 01307 Dresden, Germany. [2] Division of Theoretical Systems Biology, German Cancer Research Center, 69120 Heidelberg, Germany. [3] Faculty of Biosciences, Heidelberg University, 69120 Heidelberg, Germany. [4] DRESDEN-concept Genome Center, Center for Molecular and Cellular Bioengineering, TU Dresden, 01307 Dresden, Germany. [5] Institute for Immunology, Heidelberg University Hospital, 69120 Heidelberg, Germany. [6] Present address: Institute for Biomedical Genetics, University of Stuttgart, 70569 Stuttgart, Germany. [7] These authors contributed equally: Mina N. F. Morcos, Congxin Li, Clara M. Munz. ✉email: t.hoefer@dkfz.de; alexander.gerbaulet@tu-dresden.de

mmunophenotypic HSCs (lineage⁻ Sca-1⁺ CD117⁺ (LSK) CD48⁻/lo CD150⁺; Supplementary Fig. 1) are a heterogeneous population with regard to proliferative activity, lineage fate and repopulation capacity after transplantation. In particular, numerous HSC transplantation studies have described, to varying degrees, lineage fate biases of HSCs (reviewed in ref. [1]). The production rate of mature cells from HSCs is massively enhanced after transplantation compared to physiological conditions[2,3], and it remains unknown whether lineage differentiation pathways also differ between the two settings[4]. Recent in vivo barcoding studies of HSCs in native hematopoiesis have revealed HSC clones with different lineage output[5–8], which appear to resemble those seen after HSC transplantation[9,10]. Among them are HSCs producing only megakaryocyte progenitors (MkPs) independent of multipotent progenitors (MPPs)[7,9,10]. These findings raise the questions of the developmental relationships of distinct HSC and progenitor subpopulations in native hematopoiesis and, importantly, how distinct lineage pathways are regulated under physiological conditions to match the demand in mature cells (Fig. 1a). These questions are intimately linked with another controversy that has arisen in the context of the recent HSC fate mapping studies, suggesting steady but infrequent input from HSCs to hematopoiesis[2,11,12] or more active contribution[13–15]. Here, we measure and integrate extensive data on HSC differentiation, proliferation, and molecular heterogeneity, all obtained under physiological conditions, to address these fundamental questions. We identify a HSC source population with high expression of CD201 and Sca-1, from which increasingly committed HSCs emerge. Finally, we identify a hitherto elusive MkP subpopulation that directly links HSCs to megakaryocytes.

## Results

### Physiological hematopoiesis emerges from CD201hi Sca-1hi HSCs.

In previous HSC fate mapping studies, the actual stem cell source that supplies hematopoiesis over the lifetime of the mouse, called tip HSCs, has remained elusive[11,13–15]. The defining criterion for such tip HSCs is that after label induction, constant labeling frequency is maintained by perfect self-renewal; conversely, if an HSC subpopulation emerges from another source, its labeling frequency will approach the labeling frequency of this source with time[16]. Indeed, in existing HSC fate mapping data, a several-fold increase of HSC labeling frequency over at least one year has been observed that is fueled by a yet undefined subpopulation of tip stem cells within the phenotypic HSCs[11,13–15,17]. To selectively label HSCs for fate mapping, we generated the Fgd5ZsGreen:CreERT2/R26LSL-tdRFP mouse model (Supplementary Fig. 2a), which combines an inducible Cre expressed in HSCs[18] (Supplementary Fig. 2b) with an excision reporter with high activation threshold[19,20] (Supplementary Fig. 2c). Indeed, tamoxifen (TAM) application labeled HSCs with high specificity (Fig. 1b and Supplementary Fig. 2d–g), thus minimizing the problem of substantial concomitant labeling of progenitors downstream of HSCs seen in several studies[13–15]. Within the abundant LS⁻K and L⁻K⁻ populations, sporadic RFP⁺ cells were detected 10 days after TAM induction (Supplementary Fig. 2d, e), but such cells represented only a negligible fraction of these progenitor populations (Supplementary Fig. 2f).

To characterize HSC heterogeneity, we used surface levels of CD201 (also termed EPCR) and Sca-1, as either marker alone has previously been shown to correlate with repopulation potential of transplanted HSCs[21–25]. RFP-labeled cells were strongly enriched

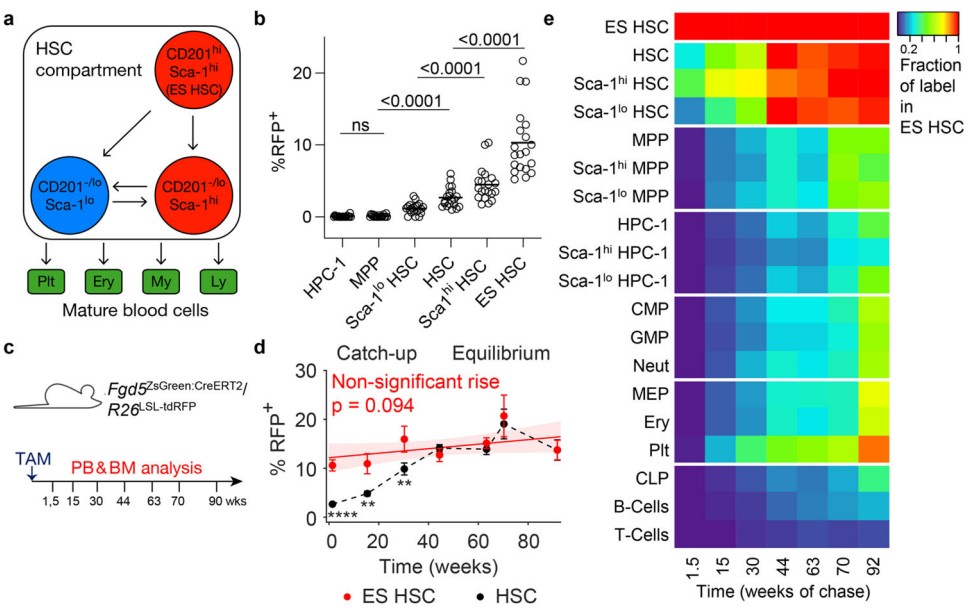

**Fig. 1 Fate mapping of CD201hi Sca-1hi HSCs. a** Subdivision of the immuno-phenotypic HSC population (LSK CD48⁻/loCD150⁺) by CD201 (EPCR) and Sca-1 expression; ES-HSC, CD201 (EPCR)hi Sca-1hi HSCs. Relations between HSC populations in native hematopoiesis and how they connect to lineage pathways will be studied. **b** Bone marrow populations of Fgd5ZsGreen:CreERT2/R26LSL-tdRFP mice were analyzed 10–11 days after TAM induction for RFP expression (n = 20, 4 independent experiments, means and individual values are shown; one-way ANOVA with Sidak correction; ns, not significant; p values shown in graph); for flow cytometry phenotypes see supplementary Fig. 1a. **c–e** To map HSC fate over time, Fgd5ZsGreen:CreERT2/R26LSL-tdRFP mice (n = 18, 10, 12, 10, 10, 8, and 12 mice/consecutive timepoint (1.5–92 weeks of chase)) were TAM-induced, groups of mice were sacrificed at indicated time points, and bone marrow populations and peripheral blood leukocytes were analyzed for RFP expression (pooled data from five independent experiments are shown). **d** Percentage of RFP⁺ HSCs and ES HSCs (mean and SEM). RFP-labeling of ES HSCs did not significantly rise (two-sided F-test on linear regression), while labeling of total HSCs did significantly increase during the catch-up phase (two-sided paired t-test, p-values for time points 1, 2, and 3: <0.001, 0.0057, and 0.0034). **e** Fractions of labeled peripheral blood and bone marrow cells relative to ES HSCs at indicated time points; the labeling frequencies of the cell populations from each individual were normalized to its ES HSC labeling at final bone marrow analysis. Means are shown. Source data for all main and supplementary figures are provided as a Source Data file. For transcriptome data, refer to "Data availability" section.

among CD201[hi] Sca-1[hi] HSCs (ES HSCs, Fig. 1b, Supplementary Fig. 2d–g). These ES HSCs were strongly enriched for quiescent cells and exhibited a primitive immuno-phenotype as judged by lower expression, compared to non-ES HSCs, of CD34[26], CD48[27], and CD117[28,29], while CD150[30] was higher expressed (Supplementary Fig. 2h).

We induced a large cohort (n = 82) of *Fgd5*[ZsGreen:CreERT2]/*R26*[LSL-tdRFP] mice to follow label propagation over time (Fig. 1c). Similar to previous studies[11,13–15], labeling frequency of the total phenotypic HSC population increased several fold over the first year (Fig. 1d, black line). By contrast, labeling of ES HSCs was initially much higher and remained approximately constant (Fig. 1d, red line, Supplementary Fig. 2i–k), which indicates perfect self-renewal of this HSC subpopulation. Moreover, the labeling frequency of all HSCs equilibrated to ES HSCs after 44 weeks of chase, implying that the latter are the ultimate source of all HSCs (Fig. 1d, e and Supplementary Fig. 2i). Indeed, ten sorted RFP[+] HSCs, but not RFP[−] HSCs, isolated from recently labeled *Fgd5*[ZsGreen:CreERT2]/*R26*[LSL-tdRFP] mice, yielded robust multi-lineage repopulation in a competitive transplantation assay (Supplementary Fig. 3a–e). In contrast, transplantation of either ten sorted RFP[+] or RFP[−] ES HSCs from recently induced *Fgd5*[ZsGreen:CreERT2]/*R26*[LSL-tdRFP] mice revealed their uniformly high repopulation potential and resulted in similar and unbiased reconstitution patterns (Supplementary Fig. 3f–m). Therefore, tracing RFP[+] ES HSCs allowed us to infer the behavior of the total ES HSC population. Taken together, these data show that ES HSCs are a self-renewing HSC population that replenish the remainder of HSCs during physiological hematopoiesis, thus placing ES HSCs upstream of other phenotypic HSCs in the hematopoietic hierarchy.

To quantify label propagation from ES HSCs through the hematopoietic system, we calculated the labeling frequencies of downstream bone marrow and peripheral blood populations relative to ES HSC labeling at final analysis of a given individual, thus normalizing for variability of initial RFP labeling frequency in ES HSCs (Supplementary Fig. 2j, k). All progenitor and mature cell populations slowly increased their labeling frequencies, trailing HSCs. Labeling of MPPs increased with considerable delay and did not equilibrate with HSC labeling within 21 months of chase (Fig. 1e), which implies that progression of cells from ES HSCs to MPPs represents a kinetic bottleneck. In accordance with previous studies[7,11,14,15,31], labeling frequency of mature cells increased fastest in platelets, slowest in T cells, and at intermediate rate in granulocytes and erythrocytes. Thus, label induced in ES HSCs reached all major progenitors and mature cells of the hematopoietic system, but with different kinetics.

**Combining fate mapping with mitotic tracking yields lineage paths.** We asked whether the detailed and time-resolved HSC fate mapping data (Fig. 1e) contain information on lineage pathways. In the case of lymphoid development, label from ES HSCs clearly arrived in progenitors before it entered mature cells (e.g., common-lymphoid progenitors (CLPs) and B cells, respectively, in Fig. 1e). Applying this principle to our data ruled out certain pathways. Specifically, classically defined megakaryocyte/erythrocyte progenitors (MEPs) could not be the sole precursors of platelets, as platelets were more strongly labeled than MEPs from 15 weeks of chase onward (Fig. 1e). We now show that the data can be used to chart lineage pathways. The topology of lineage pathways emerging from stem cells is primarily defined by branch points; it may also contain convergence points when a downstream population has more than one progenitor. Within a given pathway topology, three measurable quantities are closely related: the rates of *cell differentiation* and *proliferation* at each node

define the *size of the cell population* constituting this node (Supplementary Fig. 4a). The in vivo values of differentiation rates, proliferation rates, and population sizes can be measured by, respectively, HSC fate mapping[32], dilution of fluorescently labeled histones (e.g., H2B-GFP[33]), and cell counting (measured as percentage of lin[−] BM cells). We reasoned that obtaining these data for all cell populations in a stem cell differentiation pathway will also yield the pathway topology. To examine this, we generated a toy model for a generic branching lineage topology in silico (Supplementary Fig. 4b) and simulated fate mapping and mitotic tracking data. We then constructed all principally possible topologies linking stem cells, progenitors, and mature cells (Supplementary Fig. 4c), and asked which topology is compatible with the simulated data. Statistical model selection based on the simulated data with added noise indeed recovered the true pathway and rejected all others (Supplementary Fig. 4d, e). By contrast, using the in silico fate mapping data alone did not single out the correct model (Supplementary Fig. 4f). This example with computationally simulated data suggests that lineage pathways downstream of tip HSCs can be inferred from integrating experimental data on stem cell fate mapping, mitotic history, and cell numbers.

**The myeloid lineage diverges within phenotypic HSCs.** We applied this inference approach to our experimental fate mapping data, complementing them with measurements of H2B-GFP label dilution, and stem and progenitor cell numbers measured relative to lineage-negative (lin[−]) cells. In addition to subdivision of HSCs by CD201 and Sca-1 expression, we stratified MPPs and HPCs-1 by Sca-1 expression. This yielded a graph of all possible pathway topologies, with unidirectional differentiation steps (as implied by transplantation studies) and the possibility of reversible Sca-1 loss (Fig. 2a and Supplementary Fig. 4g, h). We systematically enumerated 144 pathways (subgraphs) that connect ES HSCs to common-myeloid progenitors (CMPs) and CLPs (Supplementary Fig. 5a) and translated these into kinetic equations describing cell abundance, RFP label propagation, and H2B-GFP dilution (Supplementary Methods, section 1). The resulting pathway models were ranked by their ability to fit the experimental data, using the bias-corrected Akaike information criterion (AICc[34]). Remarkably, a single model was substantially better than the next-ranked models (ΔAICc > 2; Fig. 2b and Supplementary Fig. 5b), showing that high Sca-1 levels were either inherited from precursor (e.g., Sca-1[hi] HSC) to progeny (e.g., Sca-1[hi] MPP) or lost irreversibly (Fig. 2c); this model fit all the experimental data (Fig. 2d–f; for parameter values see Supplementary Table 1 and Supplementary Fig. 5c). The next best ranking models (within ΔAICc < 10, indicated by orange bars in Fig. 2b; Supplementary Fig. 5b) were variants of the best model, all showing parallel Sca-1[hi] and Sca-1[lo] differentiation pathways and irreversible conversion from Sca-1[hi] to Sca-1[lo] cells at every level of the stem and progenitor cell hierarchy (Supplementary Fig. 6a, b). Models that either lacked inheritance of Sca-1 level (Fig. 2b, green bars, non-parallel) or allowed Sca-1[lo] to Sca-1[hi] transitions (Fig. 2b, blue bars, non Sca-1-high-to-low), or both (Fig. 2b, gray bars), did not account for the experimental data. These key features of model selection were recapitulated when only 50% of the experimental data set was used (Supplementary Fig. 6c), implying that the data robustly discriminate between the alternative models.

The selected model (Fig. 2c) predicts that erythro-myeloid commitment requires downregulation of Sca-1 expression either in HSCs or in progenitors (Fig. 2g). Specifically, multiple possible routes lead to CMPs. Based on the differentiation rates inferred from the experimental data, we computed the probability that a

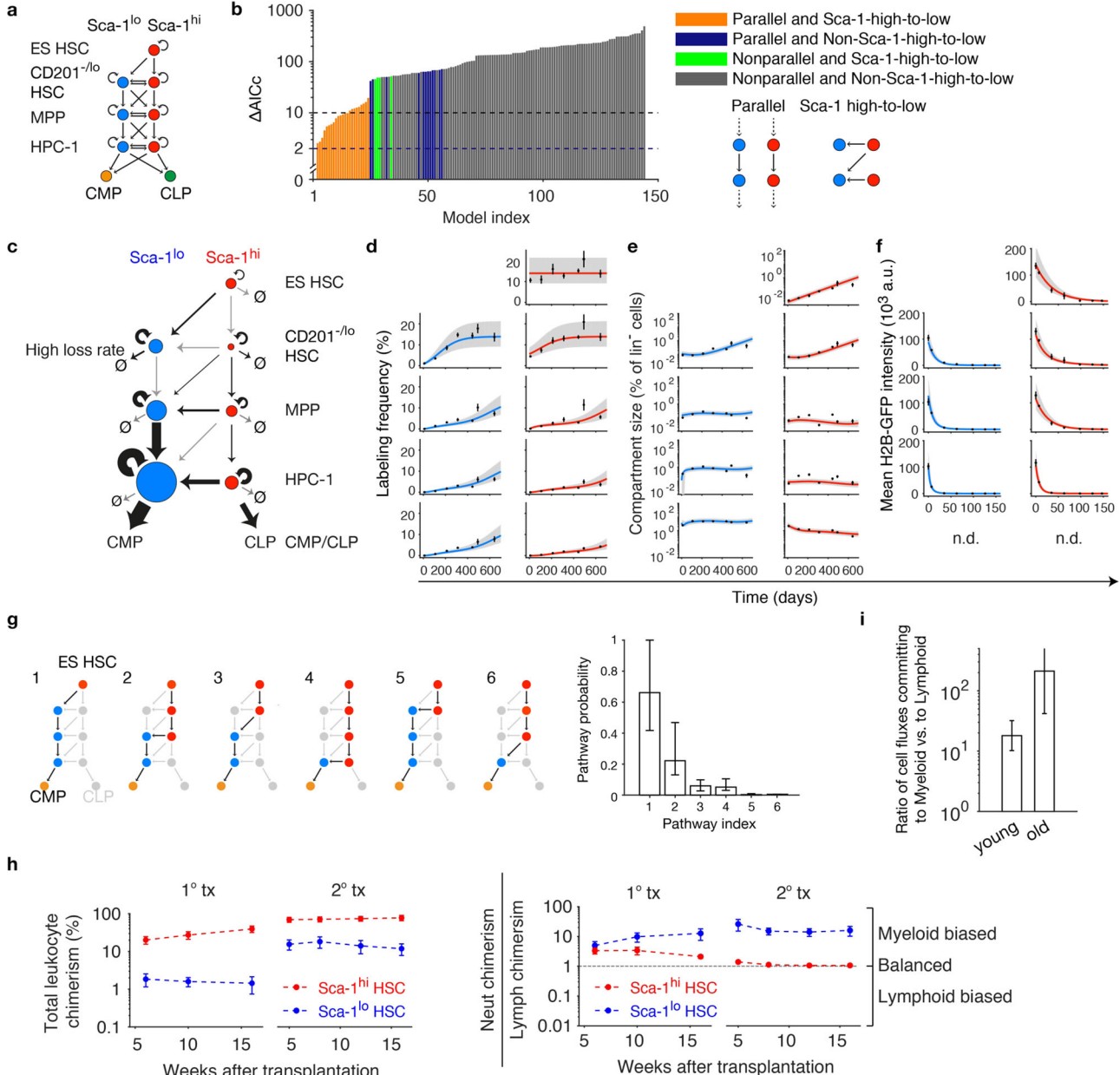

**Fig. 2 Statistical inference of HSC differentiation pathways. a** Model scheme, with subdivision of HSCs and progenitors according to Sca-1 expression and, for HSCs, CD201 expression. Straight arrows denote cell differentiation (vertical), or change in Sca-1 expression (horizontal), or both (diagonal); curved arrows denote cell proliferation. All possible differentiation are superimposed. **b** Model selection by bias-corrected Akaike information criterion (AICc) against experimental data for HSC fate mapping, mitotic history, and relative cell numbers. Models are classified (and color-coded accordingly) by two features: parallel differentiation pathways and irreversible loss of Sca-1. The best-fitting model **c** has ΔAICc = 0. None of the remaining 143 models is within ΔAICc < 2. Further models with acceptable evidence (ΔAICc < 10) are all variants of the best-fitting model (see Supplementary Fig. 5b). **c** Best model selected by experimental data. Arrow width signifies values of the corresponding differentiation or proliferation rates, while node size quantifies cell number (relative to lin⁻ cells) in each compartment at 100 days of chase; see Supplementary Table 1. Fits of the best model for RFP labeling frequency propagation (**d**), compartment size (**e**), and H2B-GFP dilution (**f**) (experimental data: mean ± SEM; mice in **d** and **e** are the same as in Fig. 1c–e; for **f**, $n = 3$, 4, 4, 2, 2, 4, 4 mice/consecutive timepoint (0−21 weeks of chase) were analyzed; model fit: straight line, best fit; gray-shaded areas, 95% confidence bounds). **g** Inferred lineage pathways to CMPs and their relative contributions to generation of CMPs (showing the probability that a CMP was generated by each pathway; error bars, 95% prediction bands). **h** As predicted by the model, transplanted Sca-1ʰⁱ HSCs, but not Sca-1ˡᵒ HSCs, yielded lymphoid progeny. 100 Sca-1ʰⁱ or Sca-1ˡᵒ donor HSCs were competitively transplanted and contribution to peripheral blood neutrophils, B and T cells was determined (data re-analyzed from Morcos et al.[25]). Lineage bias of donor-derived leukocytes was calculated (Neutrophil chimerism/[B + T cell chimerism]/2; $n = 4$ recipient mice for Sca-1ʰⁱ HSCs and $n = 3$ for Sca-1ˡᵒ HSCs in primary and secondary transplantation; mean and SEM are shown). **i** Ratio of cell fluxes between Sca-1ˡᵒ HPC-1 → CMP and Sca-1ʰⁱ HPC-1 → CLP for young and old mice (error bars, 95% prediction bands).

CMP was generated by any particular pathway (Supplementary Methods, section 2). The majority (~2/3) of CMPs could be traced back to CD201$^{-/lo}$ Sca-1$^{lo}$ HSCs (Fig. 2g, pathways 1 and 5), downstream followed by massive selective proliferation in the Sca-1$^{lo}$ pathway. Hence, in unperturbed hematopoiesis, a major commitment to erythro-myeloid differentiation occurs already at the HSC stage. This prediction is further supported by the outcome of competitive transplantation of immuno-phenotypic HSCs (LSK CD48$^{-/lo}$ CD150$^{+}$) purified according to their Sca-1 expression level, which yielded lower and preferentially myeloid reconstitution of recipients transplanted with Sca-1$^{lo}$ HSCs, while Sca-1$^{hi}$ donor HSCs resulted in durable engraftment and a balanced output (Fig. 2h).

In transplantation experiments, aged HSCs produce more myeloid than lymphoid output compared to HSCs from young mice[35]. To study whether this myeloid bias occurs during physiological hematopoiesis, we induced RFP label in HSCs of mice >1.5 years of age by TAM administration (Supplementary Fig. 7a–f), traced label propagation as described above for mice labeled as young adults (Fig. 1), and generated mitotic tracking data with H2B-GFP dilution for aged mice (Supplementary Fig. 7g–i). As previously described, aged mice exhibited increased HSC frequency[35,36] and thrombocythemia[37] (Supplementary Fig. 7b, c). The share of ES HSCs among the total HSC population was significantly larger in aged animals (Supplementary Fig. 7e), while the labeling frequency of ES HSCs in old and young animals remained constant over time (Supplementary Fig. 7d). The fundamental dynamics of label propagation from HSCs to downstream progenitor compartments were similar between aged and young mice (Supplementary Fig. 7f), except for faster label increase in total HSCs, MkPs, and platelets in aged animals.

Using the mathematical model (Fig. 2c), we inferred the differentiation, proliferation and death rates of aged HSPCs from the combined fate mapping, mitotic tracking, and cell number data, based on an accurate model fit within experimental error (Supplementary Fig. 7j). We found that the differentiation rates from CD201$^{-/lo}$ Sca-1$^{hi}$ HSCs to Sca-1$^{lo}$ MPPs and from Sca1$^{lo}$ HPC-1 to CMPs significantly increased (Supplementary Table 1), as did the respective cell fluxes (Supplementary Fig. 7k); by contrast, the faster label propagation from ES HSCs to the remainder of the HSCs (Supplementary Fig. 7k) was accounted for by the increase in ES HSC number (Supplementary Fig. 7l). As a result of these changes with aging, the flux of CMP production relative to CLP production was increased more than five-fold in aged mice, revealing an accentuated myeloid bias with aging in physiological hematopoiesis (Fig. 2i).

Finally, we noticed an extraordinarily high loss rate of CD201$^{-/lo}$ Sca-1$^{lo}$ HSCs (Fig. 2c, arrow labeled "High loss rate"), which is unlikely to be explained solely by cell death as the loss rate then would be higher than the differentiation rate to Sca-1$^{lo}$ MPPs. This puzzling observation will be addressed in the following.

**CD201$^{-/lo}$ Sca-1$^{lo}$ HSCs feed thrombopoiesis via CD48$^{-/lo}$ MkPs**. MPPs have been thought to supply all lineage pathways[38,39], but recent work indicated that platelets can be generated directly from HSCs in mice[7,9,10] and humans[40]. Indeed, when comparing labeling frequencies in committed progenitors and mature cell types relative to MPPs in our own data, only platelets showed significantly faster and higher label acquisition than both MPPs and CMPs (Fig. 3a and Supplementary Fig. 8a, b), arguing against thrombopoiesis proceeding solely via MPPs. By contrast, CD201$^{-/lo}$ Sca-1$^{lo}$ HSCs had higher labeling frequency than platelets throughout the entire chase period (Fig. 1e). These data suggest that the high loss rate of CD201$^{-/lo}$ Sca-1$^{lo}$

HSCs in the model fit (Fig. 2c) was actually caused by their yet unidentified differentiation into megakaryocyte progenitors, implying two distinct pathways of thrombopoiesis (Fig. 3b). To test this idea, we performed further fate mapping experiments that included MkPs. We found that MkPs had consistently lower RFP labeling frequencies than platelets (Fig. 3c). Thus, the fate mapping data rule out a homogeneous MkP population that is supplied by the conventional pathway via MPPs and the direct pathway from HSCs. As MkPs are well established as precursors of megakaryocytes (Mks)[41,42], we searched for solutions for this paradox. The simplest working model, as confirmed by mathematical analysis of RFP label propagation, suggests that the conventional and the direct thrombopoiesis pathways are active simultaneously and proceed through distinct MkP subpopulations (Supplementary Fig. 8c, d).

To gain unbiased insight into putative differentiation trajectories from HSCs to MkPs, we performed single-cell RNA sequencing (scRNAseq) of index-sorted HSCs, MPPs, HPCs-1, and LS$^-$K progenitors including MkPs, using Smart-seq2[43]. These populations occupied distinct but partially overlapping regions in the transcriptional landscape computed by PHATE[44] (Fig. 3d, upper left panel). The different kinds of progenitor cells were well identified by the phenotypic markers used, as indicated by preferential expression of myeloid, erythroid, and megakaryocyte lineage gene modules[3] in myeloid progenitors, erythroid progenitors, and MkPs, respectively (Fig. 3d). Moreover, Sca-1 was most highly expressed in HSCs at the apex of the transcriptional landscape (Fig. 3e), which also expressed high levels of *Vwf* and *Itga2b* (Fig. 3f) as previously described[45,46]. We noted that MkPs showed heterogeneous surface expression of CD48 (Fig. 3g), with CD48$^{-/lo}$ MkPs being significantly closer to Sca-1$^{lo}$ HSCs in transcript space (Fig. 3h). To extract putative developmental trajectories between HSC and progenitor subpopulations, we applied the graph abstraction method PAGA[47] (Fig. 3i). PAGA recovered the topology inferred from fate mapping (Fig. 3i, red lines) with high confidence (Fig. 3j). Remarkably, for thrombopoiesis, only CD48$^{-/lo}$ MkPs were directly connected to CD201$^{-/lo}$ Sca-1$^{lo}$ HSCs (Fig. 3i, green line) whereas CD48$^{hi}$ MkPs were connected to HPC-1 via LS$^-$K progenitors (Fig. 3i, blue lines). Thus, single-cell transcriptome data point to distinct origins of CD48$^{-/lo}$ and CD48$^{hi}$ MkPs. Moreover, the significant correlation of pseudotime and Mk score in each MkP subpopulation individually suggests that each subset matured in parallel in a continuous process (Fig. 3k).

PAGA also inferred a putative link between CD48$^{-/lo}$ and CD48$^{hi}$ MkPs, indicating that they could be connected. To further address the origins of the MkP subpopulations, we performed in vivo fate mapping. Indeed, fate mapping label was rapidly acquired by CD48$^{-/lo}$ MkPs in induced *Fgd5*$^{ZsGreen:CreERT2}$/ *R26*$^{LSL-tdRFP}$ mice, whereas CD48$^{hi}$ MkPs acquired less label at slower kinetics (Fig. 4a, b). The CD48$^{-/lo}$ MkP labeling quickly exceeded its own initial labeling as well as of all other progenitors, suggesting that CD48$^{-/lo}$ MkPs directly acquire label from HSCs. Consistent with pseudotime analysis (Fig. 3k), the complete lack of label equilibration between the two MkP subsets implies that they are located in two distinct pathways (rather than CD48$^{-/lo}$ MkPs being upstream of CD48$^{hi}$ MkPs). The LSK CD48$^{hi}$ CD150$^{+}$ population (HPC-2), which has been implied in thrombopoiesis[7,15,39], could not be upstream of CD48$^{-/lo}$ MkPs as evidenced by their consistently lower RFP labeling, but might give rise to CD48$^{hi}$ MkPs (Supplementary Fig. 8e). As predicted by PAGA, CD48$^{-/lo}$ MkPs were developmentally close to HSCs, as they were the only cell type among all LS$^-$K progenitors which retained or inherited significant amounts of H2B-GFP in chased *R26*$^{rtTA}$/*Col1A1*$^{H2B-GFP}$ mice (Fig. 4c). Functional characterization of both MkPs by culture and transplantation (Fig. 4d, e and

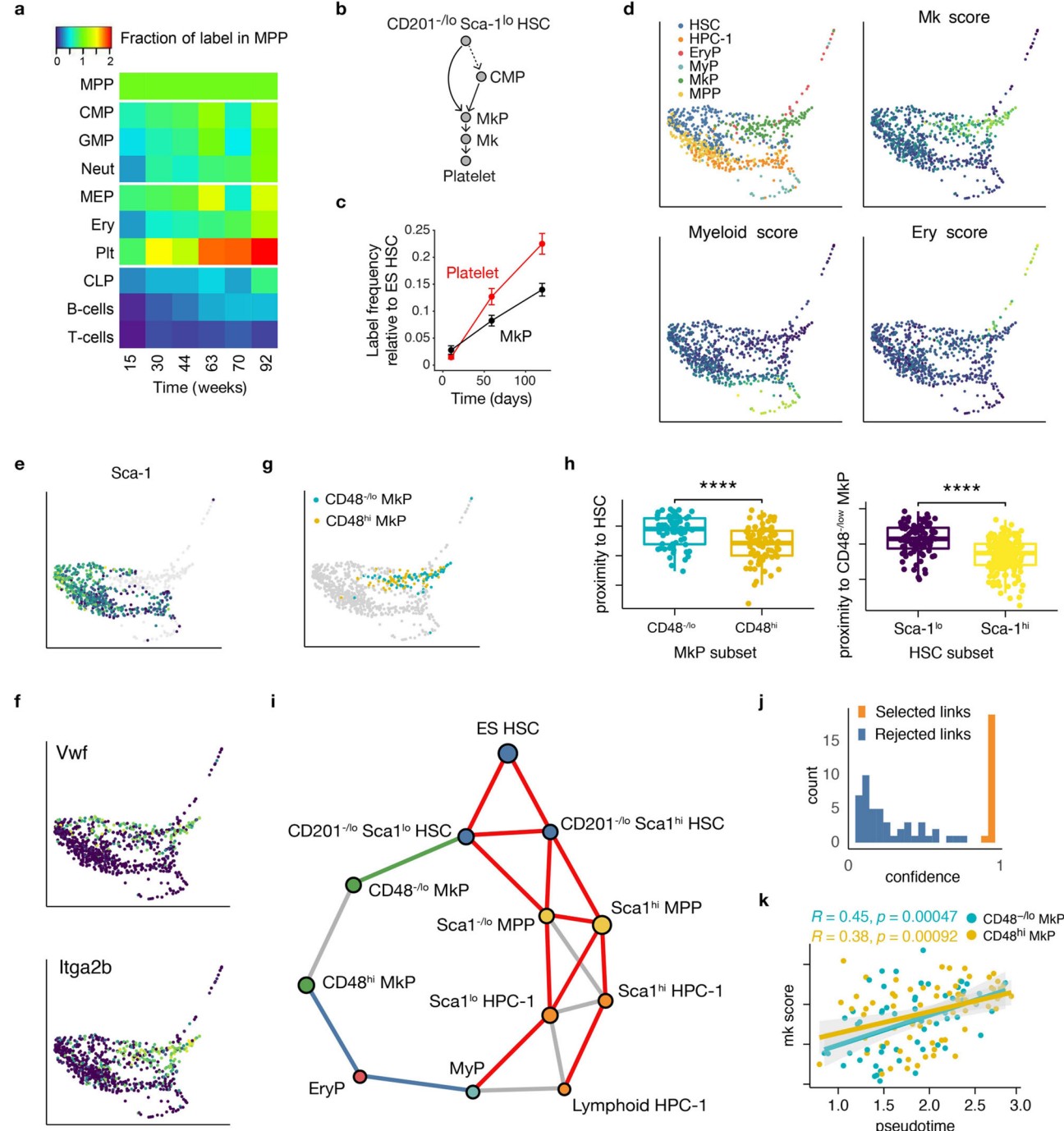

Supplementary Fig. 8f–h) confirmed their predominant thrombocytic potential with some residual oligopotent behavior. While most cultivated CD48$^{hi}$ MkP gave rise to a single Mk, significantly more CD48$^{-/lo}$ MkP proliferated before acquiring a megakaryocytic morphology. Likewise, transplantation demonstrated enhanced platelet generation from CD48$^{-/lo}$ MkPs. While both MkP populations similarly expressed megakaryocytic markers (Supplementary Fig. 8i–k), CD48$^{hi}$ MkPs expressed higher levels of myeloid genes and CD48$^{-/lo}$ MkP harbored an erythroid signature (Supplementary Fig. 8j–k), which was further evidenced by their robust erythroid potential upon transplantation (Fig. 4e).

To summarize, combining the data from RNAseq, fate mapping, and mitotic tracking, we identified a pathway of thrombopoiesis proceeding directly from CD201$^{-/lo}$ Sca-1$^{lo}$ HSCs to CD48$^{-/lo}$ MkPs that is used in parallel with the conventional pathway of platelet generation via MPPs and CD48$^{hi}$ MkPs (Fig. 4f).

Modeling label propagation through these pathways now gave excellent agreement with the experimental data (Fig. 4g). We used the model to infer how many megakaryocytes are produced per day through these distinct pathways (Fig. 4h). These pathway fluxes are the product of differentiation rate (which is higher for CD48$^{-/lo}$ MkPs, consistent with transplantation outcome (cf. Fig. 4e)) and precursor number (which is higher for CD48$^{hi}$ MkPs), resulting in equal contributions of both pathways to steady-state thrombopoiesis in young mice (Fig. 4f–i). The higher differentiation rate of CD48$^{-/lo}$ MkPs, compared to CD48$^{hi}$ MkPs is sustained by both their own proliferation and influx from CD201$^{-/lo}$ Sca-1$^{lo}$ HSCs (Supplementary Table 1). Moreover, the

**Fig. 3 Direct and conventional thrombopoiesis pathways proceed via distinct progenitors. a** RFP labeling of cell populations from $Fgd5^{\text{ZsGreen:CreERT2}}/R26^{\text{LSL-tdRFP}}$ mice (same as in Fig. 1c–e) shown as fraction of label in the MPPs of the same animal at final analysis. **b** Model scheme with direct route to MkPs (LS⁻K CD41⁺CD150⁺) from CD201⁻/lo Sca-1lo HSCs. **c** $Fgd5^{\text{ZsGreen:CreERT2}}/R26^{\text{LSL-tdRFP}}$ mice were TAM-induced and indicated cell populations were analyzed at 10, 59, and 120 days ($n = 5$ animals/ time point, means and SEM are shown). **d** PHATE embedding of single-cell transcriptomes of index-sorted hematopoieitic stem and progenitor cells ($n = 820$ cells total) isolated from 6 $R26^{\text{rtTA}}/Col1A1^{\text{H2B-GFP}}$ animals, colored by cell type and expression of lineage markers (blue: low, yellow: high). **e** HSCs, MPPs, and HPCs-1 show a decrease of Sca-1 surface expression in proximity of MkPs and myeloid-committed progenitors. **f** PHATE plots showing *Vwf* and *Itga2b* expression. **g** Lower surface expression of CD48 in MkPs residing closer to HSCs in the transcriptional landscape. **h** Proximity in transcriptional space of MkP subsets to HSCs (left), and of HSC subsets to CD48⁻/lo MkPs (right), placing CD48⁻/lo MkPs close to Sca-1lo HSCs. (two-sided Wilcoxon test; left plot: $n = 135$ cells, right plot: $n = 279$ cells; p-value ≤ $10^{-4}$ for both comparisons; boxplot description: central line indicates median, upper (lower) hinges indicate third (first) quartiles, top (bottom) whisker extends to largest value within a 1.5× inter-quantile range from top (bottom) hinge). **i** PAGA embedding of surface marker-defined populations, displaying transcriptional proximity of CD201⁻/lo Sca-1lo HSC to CD48⁻/lo MkP (green link), and a distinct differentiation path to CD48hi MkP (blue) via myeloid (MyP, LS⁻K CD41⁻ CD150⁻) and erythroid progenitors (EryP, LS⁻K CD41⁻CD150⁺). Red PAGA links were previously inferred by statistical model selection (see Fig. 2c). Cell populations were identified via indexed surface markers, except for lymphoid-biased HPCs-1, which were identified by their transcriptional signature. **j** Confidence score for PAGA links. **k** Comparison between Mk score and pseudotime shows parallel Mk maturation trajectories in CD48⁻/lo and CD48hi MkP subsets. Gray areas represent 95% confidence bounds for linear regression, R indicates the Pearson correlation coefficient, p-value is relative to a two-sided alternative hypothesis.

differentiation flux of MPPs to CD48hi MkPs, via heterogeneous CMPs, was minor compared to MPP contribution to erythropoiesis and myelopoiesis (Supplementary Fig. 9a–g). Additionally, the small fraction of initially labeled MPPs and MkPs did not have a significant impact on label propagation in the model, supporting that HSCs are the relevant source of fate-mapping label for all assayed lineages (Supplementary Fig. 9h).

The faster differentiation of CD48⁻/lo MkPs also implies that this MkP subset more efficiently gives rise to megakaryocytes and suggests that it could be utilized preferentially upon increased demand for platelets. Mice exhibit an approximately 1.5–2-fold increase in platelet count during aging[46,48,49]. Indeed, our HSC fate mapping in aged mice (Supplementary Fig. 7) revealed an increased contribution of the CD48⁻/lo MkP pathway, accounting for about 80% of thrombopoiesis (Fig. 4i). Consistently, we observed increased platelet numbers in aged mice (Supplementary Fig. 7c).

**Thrombopoietin signaling enhances direct thrombopoiesis.** The differentiation rate of HSCs is thought to respond to the need for mature cell types after injury or other challenges, but a direct assay for quantifying HSC differentiation in physiological hematopoiesis has been lacking. We reasoned that HSC fate mapping may provide such an assay. To examine this, we exposed TAM-induced $Fgd5^{\text{ZsGreen:CreERT2}}/R26^{\text{LSL-tdRFP}}$ and DOX-pulsed $R26^{\text{rtTA}}/Col1A1^{\text{H2B-GFP}}$ mice to the myeloablative drug 5-Fluorouracil (5-FU), and investigated how ES HSCs contribute to hematopoietic recovery (Fig. 5a, b). 5-FU substantially accelerated the contribution of RFP-labeled HSCs to progenitors and mature cells (Fig. 5c and Supplementary Fig. 10a), consistent with previously published data[8,11]. Along with increased HSC output, 5-FU provoked H2B-GFP dilution equivalent to ~3–4 additional cell divisions in all HSC and MPP subpopulations (Supplementary Fig. 10b, c). In CD48hi MkPs, 5-FU diluted the H2B-GFP label to the range of the background control. By contrast, CD48⁻/lo MkPs still exhibited H2B-GFP labeling after recovery from 5-FU treatment, which further supports their direct emergence from HSCs. Taken together, these experiments show that HSC fate mapping provides an assay for changes in HSC output in response to perturbation.

We used this assay to study the effects of thrombopoietin signaling, which is a physiological regulator of thrombopoiesis as well as HSC maintenance. To this end, we treated $Fgd5^{\text{ZsGreen:CreERT2}}/R26^{\text{LSL-tdRFP}}$ and $R26^{\text{rtTA}}/Col1A1^{\text{H2B-GFP}}$ mice with the thrombopoietin receptor (Mpl) agonist, romiplostim, for 5 consecutive days (Fig. 5d, and Supplementary Fig. 10a, d–f). As previously reported[50], this regimen expanded all LSK

subpopulations except MPPs (Supplementary Fig. 10f) and raised platelet counts (Fig. 5e). To understand how this is achieved, we studied label propagation from ES HSCs to platelets. RFP labeling frequencies of total HSCs including ES HSCs were not affected by romiplostim treatment (Fig. 5f–g). However, we observed strong increases in RFP-labeling frequencies in downstream progenitors, including MPPs and CD48⁻/lo MkPs, implying accelerated differentiation of HSCs into these progenitors (Fig. 5g and Supplementary Fig. 10a). In addition, romiplostim administration to $R26^{\text{rtTA}}/Col1A1^{\text{H2B-GFP}}$ mice induced ~1–2 additional divisions in HSCs and progenitors (Supplementary Fig. 10d, e). To understand how the acceleration of differentiation and enhanced cell division conspire to increase platelet production, we fitted the mathematical model to the experimental data (Fig. 4f and Supplementary Fig. 11a–c). The flux through the direct pathway was increased at all stages from ES HSCs to megakaryocytes, resulting in an almost 10-fold increase of the pathway flux (Fig. 5h). By comparison, the flux through the conventional pathway was only moderately upregulated (Fig. 5h, i). Taken together, thrombopoietin signaling enhances platelet production by channeling CD201⁻/lo Sca-1lo HSCs into the direct thrombopoiesis pathway (Fig. 5j).

## Discussion

We have demonstrated that stem cell fate mapping charts physiological differentiation pathways when combined with proliferation assays and mathematical inference. We thus pinpoint tip stem cells of native hematopoiesis and show how lineage pathways diverge immediately downstream of tip HSCs. Specifically, we find that thrombopoiesis proceeds via two distinct progenitors that previously were thought to be a homogeneous population. Our identification of MkP subpopulations has been enabled by closely linking single-cell transcriptomics with fate mapping and mitotic tracking. Moreover, the accuracy of predictions derived from computational model selection attest to the utility of this tool for interrogating complex, multi-faceted data. Our results on HSC self-renewal and differentiation rates are consistent with previous studies employing a different Cre driver ($Tie2^{\text{MeriCreMer}}$) to label HSCs for fate mapping[11,17]. Surface markers of stem and progenitor cells have been selected for their utility in prospective isolation of these cells, and less emphasis has been given to their molecular function. Interestingly, however, Sca-1, marking tip-HSCs together with CD201, is an interferon-stimulated gene (ISG)[51]. Constitutive expression of ISGs, with protective function, is a hallmark of stem cells also beyond the hematopoietic system[52]. CD201 has been implied in HSC retention to the niche[53]. Our functional data on tip HSCs imply that

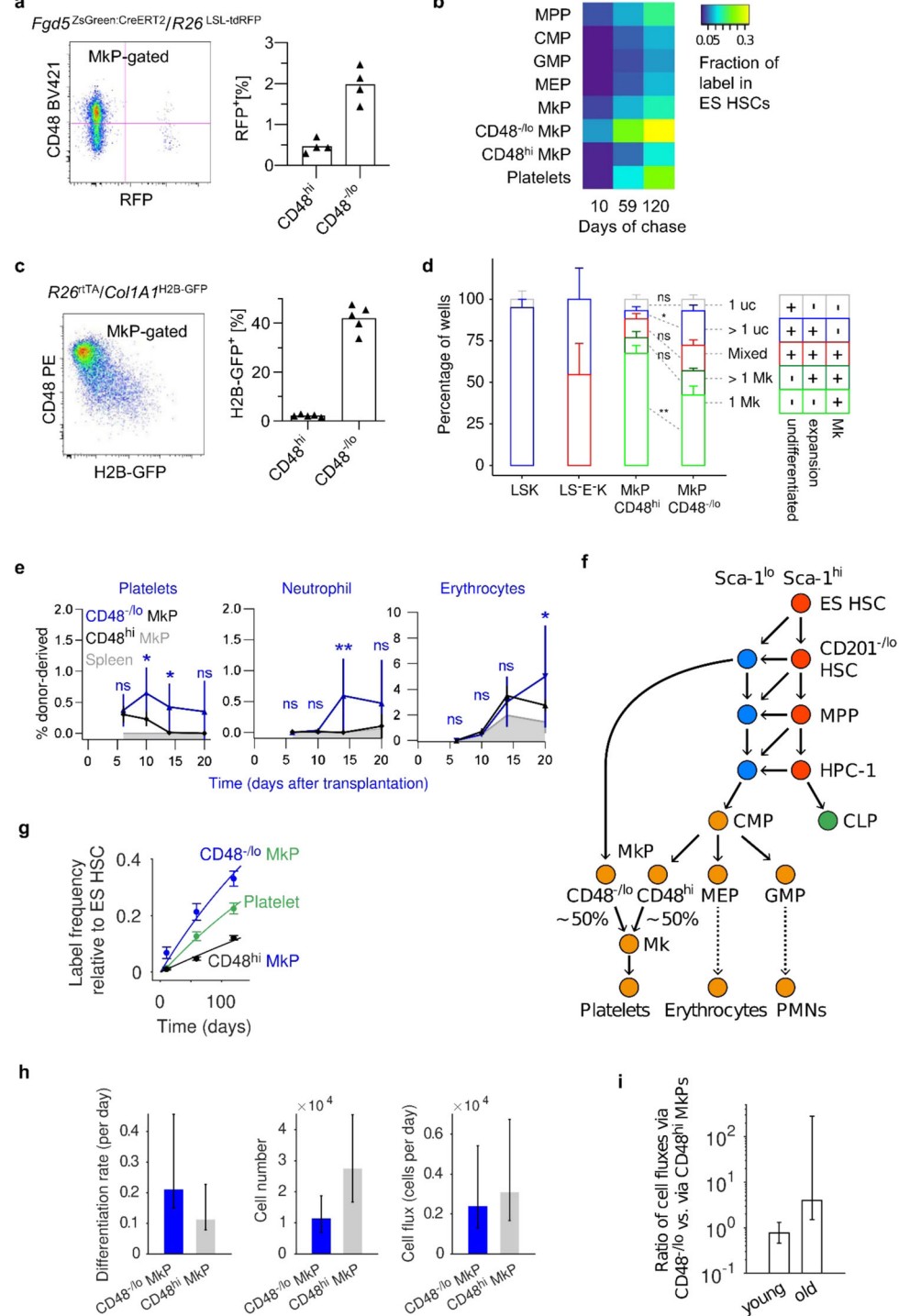

the gene expression program associated with high co-expression of Sca-1 and CD201 deserves closer scrutiny.

Our identification of a short pathway of thrombopoiesis, leading from Sca-1$^{lo}$ HSCs via CD48$^{-/lo}$ MkPs to megakaryocytes, strengthens the recently advanced view that thrombopoiesis is special among all adult hematopoietic lineage pathways by emerging directly from HSCs, and provides a specific progenitor for this pathway[7,9,40]. Contrary to the classical view[15,38], but in agreement with Rodriguez-Fraticelli et al.[7], this pathway does not proceed via MPP. Our fate mapping and single cell transcriptome data uncovered that each MkP subset matured independently and that the cell flux of CD48$^{-/lo}$ MkP was accelerated as this two-fold

smaller population had higher proliferative potential and contributed equally to platelet production.

While platelets, neutrophils, and erythrocytes make up the major share of hematopoietic cell production in steady state, their amplification strategies are fundamentally different: Neutrophil and erythrocyte numbers are achieved by massive progenitor proliferation, whereas platelet numbers are obtained via prolonged shedding from megakaryocytes after enormous cell growth. This special amplification strategy is linked to the evolution of anucleated platelets in mammalian hematopoiesis[54]. We now find that these two strategies for amplifying cell numbers are already manifest at the HSC level. The major cell flux from Sca-

**Fig. 4 Functional characterization of CD48$^{-/lo}$ and CD48$^{hi}$ MkP subpopulations. a, b** $Fgd5^{ZsGreen:CreERT2}/R26^{LSL-tdRFP}$ mice were TAM-induced, and indicated cell populations were analyzed at 10, 59, and 120 days ($n = 4$ mice/time point). **a** Representative example of enriched RFP labeling in CD48$^{-/lo}$ MkPs 59 days after TAM induction. Quantification of label in MkP subpopulations (right data plot, individuals, and means are shown). **b** RFP labeling of cell populations displayed as fraction of ES HSCs. **c** $R26^{rtTA}/Col1A1^{H2B-GFP}$ animals ($n = 5$) were DOX-induced and chased for 11 wks. Representative dot plot (left) and frequency of label retaining (H2B-GFP$^+$) MkP subpopulations (right data plot, individual mice, and means are shown). **d** Single CD48$^{-/lo}$ ($n = 137$) and CD48$^{hi}$ ($n = 122$) MkPs as well as control cells (LS−E−K: lin− CD201− Sca-1− CD117$^+$ progenitors ($n = 17$); LSK: lin− Sca-1$^+$ CD117$^+$ cells ($n = 23$)) were cultivated for 5 days and classified according to expansion and morphology. Bars represent mean values, and error bars SEMs. **e** 2500 CD48$^{hi}$ or CD48$^{-/lo}$ MkPs were purified from B6.RFP donor animals and transplanted into sublethally-irradiated B6.CD45.1/CD45.2 recipient mice ($n = 7$/condition). Contribution to platelets, erythrocytes, neutrophils was monitored for 3 weeks (mean and SD are shown). Transplanted LSK cells served as positive control ($n = 1$, see Supplementary Fig. 8h), transplanted B6.CD45.1/.2 carrier splenocytes as negative controls ($n = 3$); significance calculated with repeated measures 2-way ANOVA with Holm–Sidak error correction (ns, not significant; *, <0.05; **, <0.01). **f** Full model of thrombopoiesis. The direct (via CD48$^{-/lo}$ MkP) and the conventional (via CMP-CD48$^{hi}$ MkP) pathways equally contribute to native thrombopoiesis. **g** Mathematical inference on the fate mapping data based on the two-pathway model of thrombopoiesis fits the experimental data ($n = 4$ mice/time point; same mice as in **a** and **b**, mean and SEM are shown) and confirms the model prediction that MkPs consist of two distinct subpopulations (see Supplementary Fig. 8c, d). **h** CD48$^{-/lo}$ MkPs differentiate more frequently than CD48$^{hi}$ MkPs (left plot), while the cell number is lower in CD48$^{-/lo}$ MkP (middle plot). Overall differentiating cell fluxes (number of cells becoming Mk per day) from both MkP subpopulations are balanced (right plot). **i** Contributions of the two megakaryopoiesis pathways are balanced in young adult mice, whereas CD48$^{-/lo}$ MkPs supply more Mks in aged mice. In **h** and **i** error bars represent 95% confidence bounds.

1$^{lo}$ HSCs supplies thrombopoiesis directly, while myelopoiesis is fed by a much smaller flux from Sca-1$^{lo}$ HSCs and amplified by selective proliferation of Sca-1$^{lo}$ progenitors downstream. Moreover, the short thrombopoiesis pathway can produce platelets more rapidly than the conventional pathway, as CD48$^{-/lo}$ MkPs differentiate faster into megakaryocytes than their CD48$^{hi}$ counterparts. This feature may explain the higher flux through the short pathway under conditions of increased demand for platelet production during aging and upon applying a thrombopoietin receptor agonist. These findings would be consistent with the view that the conventional pathway may reflect a residual ability of myeloid progenitors to give rise to platelets. This is further supported by our finding that the differentiation flux from heterogeneous CMPs to myeloid and erythroid lineages is much larger than to the conventional thrombopoiesis pathways.

Our observation that activating a physiological regulator of thrombopoiesis further accentuates this effect (increasing the flux from Sca-1$^{lo}$ HSCs to thrombopoiesis and accelerating, in a compensatory manner, proliferation of Sca-1$^{lo}$ MPPs) strengthens this interpretation. Thus, our data show that HSCs contribute very actively to thrombopoiesis in native hematopoiesis, whereas they feed only infrequently into myelopoiesis, which is sustained primarily by progenitor amplification. Accordingly, mice with a severe reduction of HSCs feature normal numbers of leukocyte and erythrocytes, but a significant reduction of platelets[55,56].

In sum, the approach developed here connects the charting of lineage pathways with the quantification of cell fluxes. It, therefore, has potential for the quantitative understanding of the regulation of native hematopoiesis in steady state and in response to challenges.

## Methods

**Mice**. All animal experiments were conducted according to institutional guidelines and in accordance with the German Law for Protection of Animals approved by Landesdirektion Dresden (TVV 91/2017 and TVA24/2017). Male and female mice were used for experiments and housed in individually ventilated cages under specific-pathogen free environment at the Experimental Center of the Medical Faculty, TU Dresden. $Fgd5^{ZsGreen:CreERT2/wt}/R26^{LSL-tdRFP/LSL-tdRFP}$[18,19], $R26^{rtTA/rtTA}/Col1A1^{H2B-GFP/H2B-GFP}$[57] (Jax No: 016836, backcrossed for three generations to C57Bl/6J mice upon receipt), C57Bl/6JRj wt (Janvier), B6.CD45.1 (Jax No. 002014) and B6.CD45.1/.2 mice were used in this study. Unless stated otherwise, mice were 8-20 weeks of age when experiments were started. B6.RFP mice with ubiquitous tdRFP expression were generated by germline excision of the loxP flanked STOP cassette (LSL) in $R26^{LSL-tdRFP}$ animals[19] employing the pgk-Cre transgene[58]. Pulse-chase data of unperturbed $R26^{rtTA}/Col1A1^{H2B-GFP}$ mice was previously published[33]. H2B-GFP mice were induced with doxycyclin (DOX) via chow (Ssniff Spezialdiäten), 625 mg/kg for 10–11 weeks or 2000 mg/kg for either 2–7 weeks (this study) ad libitum. $Fgd5^{ZsGreen:CreERT2}/R26^{LSL-tdRFP}$ mice were induced by oral gavage of TAM (0.2 mg/g BW) twice 3–4 days apart. The Mpl agonist romiplostim (Nplate, Amgen, 2.5 µg/mouse/d) was injected intraperitoneal (i.p.) for 5 consecutive days. 5-FU (150 µg/g body weight, Applichem) was administered via intravenous (i.v.) injection.

**Cell Preparation**. Whole bone marrow cells were isolated by crushing long bones with mortar and pestle using PBS/2% FCS/2 mM EDTA and filtered through a 100 µm mesh. After erythrocyte lysis in hypotonic NH$_4$Cl-buffer, cells were filtered through a 30 µm mesh. Hematopoietic lineage$^+$ cells were removed with the lineage cell depletion kit (Miltenyi Biotec).

Peripheral blood was drawn into glass capillaries by retro-bulbar puncture. For identification of RFP$^+$ platelets and erythrocytes, 1–2 µl of heparinized blood was mixed with PBS/2%FCS/2 mM EDTA and incubated with monoclonal antibodies against CD41 and Ter119 (for gating see Supplementary Fig. 1c). For leukocyte analysis, erythrocyte lysis in hypotonic NH4Cl-buffer was performed twice for 5 min and cells were stained with monoclonal antibodies (Supplementary Table 2). For hemograms, blood was drawn by retro-bulbar puncture directly into EDTA-coated tubes (Sarstedt) and analyzed on a XT-2000i Vet analyzer (Sysmex).

**Transplantation**. B6.CD45.1/.2 recipient mice received a single dose of 9 Gray total body irradiation (Yxlon Maxi Shot γ-source). Test donor cells were sorted, mixed with 200000 B6.CD45.1 competitor bone marrow cells and administered via intravenous injection into the retro-orbital sinus. Peripheral blood T-lymphocytes (CD3$^+$), B-lymphocytes (B220$^+$), and neutrophils (CD11b$^+$, Gr-1$^{hi}$) were analyzed for their donor origin using an Aria III flow cytometer. The frequency of HSCs with long-term multi-lineage repopulation potential was estimated by ELDA[59]. Competitive transplantation of either Sca-1$^{hi}$ or Sca-1$^{lo}$ donor HSCs was previously described[25]. Briefly, 100 test donor HSCs (LSK CD48$^{-/lo}$ CD150$^+$) from WT B6 mice were sorted for Sca-1 expression level, mixed with 500,000 B6.CD45.1 competitor bone marrow cells and i.v. injected into irradiated B6.CD45.1/.2 recipient mice.

For transplantation of MkP populations, 2500 either CD48$^{-/lo}$ or CD48$^{hi}$ MkPs were purified from B6.RFP donor mice, mixed with 500,000 B6.CD45.1/.2 carrier splenocytes and injected into B6.CD45.1/.2 recipient mice that previously received 4 Gy γ-irradiation. Recipient mice that were transplanted with 2500 lin−Sca-1$^+$ CD117$^+$ (LSK) cells served as positive controls, while animals that received only carrier splenocytes were used as background controls.

**Flow cytometry**. Cell suspension were incubated with antibodies (Supplementary Table 2) in PBS/2% FCS/2 mM EDTA for 30 min, washed twice and analyzed on either FACS Canto, ARIA II SORP, Aria II, ARIA III (all from BD Biosciences and operating with BD FACSDiva software versions 6–8, Heidelberg, Germany) or MACSquant (Miltenyi) flow cytometers. Data were analyzed with FlowJo V9.9 and V10 software (Tree Star) and gates were set with the help of Fluorescence-Minus-One controls. For bone marrow cell surface phenotype analysis, a cocktail of biotinylated antibodies against hematopoietic lineage markers (anti-CD3e (clone 145-2C11, 1:400), anti-CD4 (clone GK1.5, 1:400), anti-CD8a (clone 53−6.7, 1:800), anti-CD11b (clone M1/70, 1:800), anti-CD19 (clone ebio1D3, 1:400), anti-LY-6C/G (Gr-1) (clone RA3-8C5, 1:400), anti-NK1.1 (clone PK136, 1:800) from eBioscience, and anti-CD45R (B220) (clone RA3-6B2, 1:400), anti-Ter119 (clone TER-119, 1:400) from Biolegend) was used, followed by a secondary staining with either 1. anti-Streptavidin (Horizon V500, 1:800, BD), anti-CD117(c-kit) (APC/eF780, clone 2B8, 1:1600, eBioscience), anti-Ly-6A/E (Sca-1) (PCP/Cy5.5, clone D7, 1:400, eBioscience), anti-CD150 (PE/Cy7, clone TC15-12F12.2, 1:200, Biolegend), anti-CD48 (BV421, clone HM48-1, 1:400, BD), anti-CD201 (EPCR) (APC,

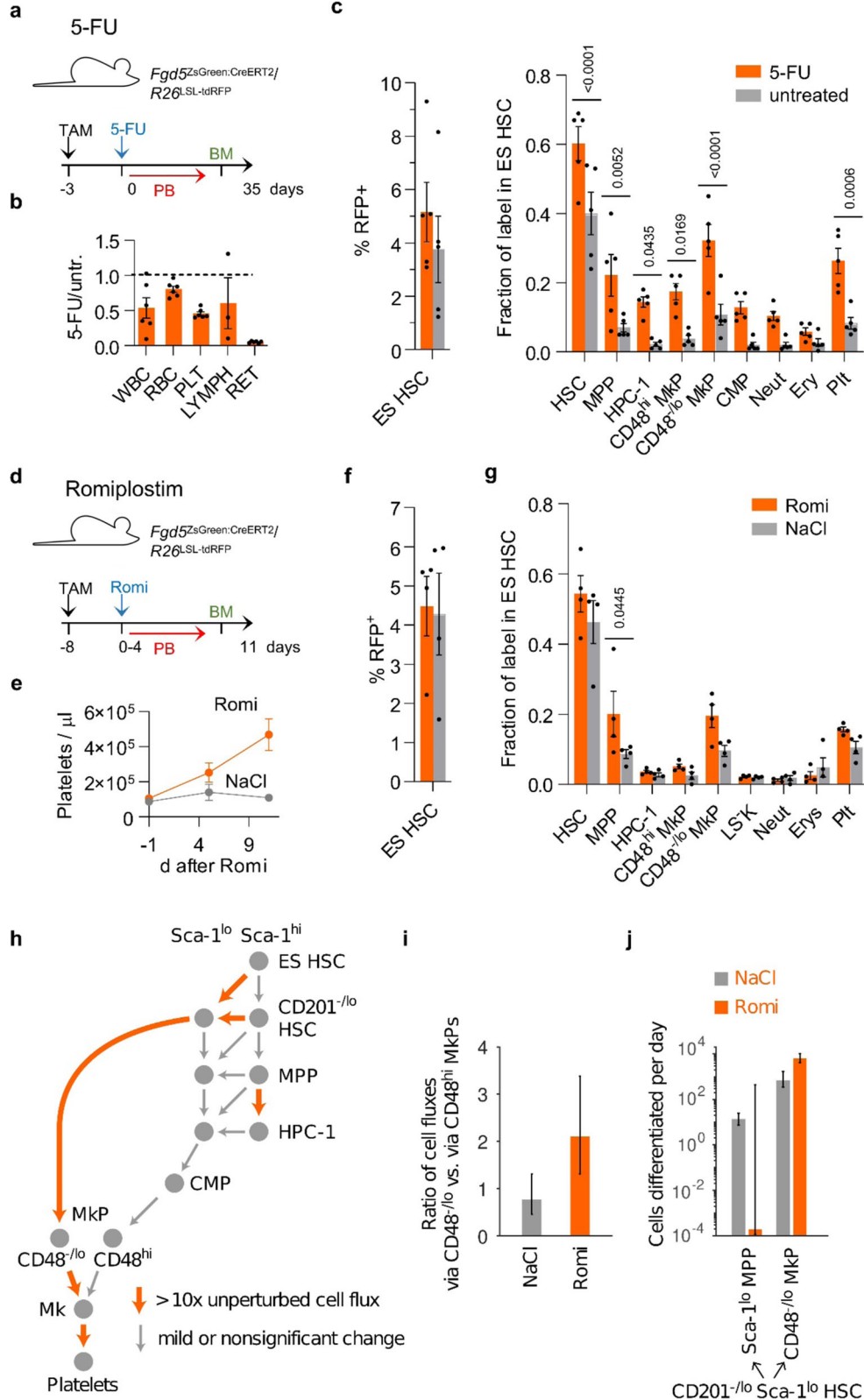

clone ebio1560, 1:100, eBioscience) and anti-CD41 (BV605, clone MWReg30, 1:100, Biolegend) for HSC and early progenitor analysis in Fgd5$^{ZsGreen:CreERT2}$/R26$^{LSL-tdRFP}$ -animals or 2. anti-Streptavidin (Horizon V500, 1:800, BD), anti-CD117(c-kit) (APC/eF780, clone 2B8, 1:1600, eBioscience), anti-Ly-6A/E (Sca-1) (PCP/Cy5.5, clone D7, 1:400, eBioscience), anti-CD34 (eF660, clone RAM34, 1:25, eBioscience), anti-CD127 (PE/Cy7, clone A7R34, 1:50, Biolegend), anti-CD135 (BV421, clone A2F10, 1:100, Biolegend) and anti-CD16/32 (AF700, clone 93, 1:100, eBioscience) for committed progenitor (CMP, GMP, MEP, CLP) analysis in

Fgd5$^{ZsGreen:CreERT2}$/R26$^{LSL-tdRFP}$-animals, or 3. anti-Streptavidin (eF710, 1:200, eBioscience), anti-CD117(c-kit) (APC/eF780, clone 2B8, 1:1600, eBioscience), anti-Ly-6A/E (Sca-1) (PCP/Cy5.5, clone D7, 1:400, eBioscience), anti-CD150 (PE/Cy7, clone TC15-12F12.2, 1:200, Biolegend), anti-CD48 (PE, clone HM48-1, 1:400, eBioscience), anti-CD201 (EPCR) (APC, clone ebio1560, 1:100, eBioscience), anti-CD34 (eF450, clone RAM34, 1:25, eBioscience) and anti-CD41 (BV605, clone MWReg30, 1:100, Biolegend) for R26$^{rtTA}$/Col1A1$^{H2B-GFP}$ -animals was performed.

**Fig. 5 Fate mapping of perturbed hematopoiesis. a–c** $Fgd5^{ZsGreen:CreERT2}/R26^{LSL-tdRFP}$ mice were TAM-induced, treated with 5-FU ($n = 5$) or left untreated ($n = 5$), sacrificed at 35 days after treatment, and bone marrow populations and peripheral blood leukocytes were analyzed for RFP expression. **b** Ratios of peripheral blood cell numbers in treated (5-FU) vs. untreated (untr.) animals 6 days after treatment (means and SD are shown). **c** Label in populations 35 days after treatment is shown as percentage of RFP$^+$ cells in ES HSC (left graph) or normalized as fraction of label in ES HSC (right graph, mean with SEM, one-way ANOVA with Sidak correction). **d–g** $Fgd5^{ZsGreen:CreERT2}/R26^{LSL-tdRFP}$ mice were TAM-induced, treated with romiplostim ($n = 4$) or saline (NaCl, $n = 4$), and sacrificed 11 d after start of treatment and hematopoietic populations were analyzed for RFP expression. **e** Time-course of platelet numbers (means and SD are shown). **f** RFP labeling in ES HSCs (means and SEM are shown). **g** Label in bone marrow and blood populations is shown as fraction of label in ES HSC in respective populations (mean with SEM, one-way ANOVA with Sidak correction). **h** Graphical overview of altered cell fluxes (number of cells differentiated per day) by romiplostim application. Orange arrows represent >10× upregulated cell fluxes. Gray arrows indicate minor or non-significant changes. (cf. Supplementary Fig. 11). **i** Inferred ratio of cell fluxes via CD48$^{-/lo}$ MkP to cell fluxes via CD48$^{hi}$ MkP at steady state (NaCl, gray) or after romiplostim (orange) perturbation. **j** Cell fluxes from CD201$^{-/lo}$ Sca-1$^{lo}$ HSCs into Sca-1$^{lo}$ MPPs and CD48$^{-/lo}$ MkPs at steady state (NaCl, gray) or after romiplostim (orange) perturbation. In **i** and **j**, the error bars represent 95% confidence bounds.

For cell surface phenotype analysis of peripheral blood cells, cells either underwent erythrocyte lysis and were stained with anti-CD3 (APC, clone 145-2C11, 1:100, or PCP/eF710, clone eBio500A2, 1:400;), anti- CD11b (APC/eF780, clone M1/70, 1:600, or FITC, clone M1/70, 1:400), anti-CD19 (PE/Cy7, clone eBio1D3, 1:200), and anti- Ly-6C/G (Gr1) (eF450, clone RB6-8C5, 1:1000, or PCP/Cy5.5, clone RB6-8C5, 1:1200), all from eBioscience, or were stained directly with anti-CD41 (APC, clone eBioMWReg30, 1:50, eBioscience) and anti-Ter119 (FITC, clone TER-119, 1:100, Biolegend).

Peripheral blood cells of transplanted B6.CD45.1/.2 mice were stained with anti-CD3 (PCP/eF710, clone eBio500A2, 1:400), anti-CD11b (APC/eF780, clone M1/70, 1:600), anti-CD19 (PE/Cy7, clone eBio1D3, 1:200), and anti- Ly-6C/G (Gr1) (eF450, clone RB6-8C5, 1:1000) from eBioscience and anti-CD45.1 (APC, clone A20, 1:800) and anti-CD45.2 (FITC, clone 104, 1:400) from Biolegend.

Bone marrow cells of transplanted B6.CD45.1/.2 mice were stained with anti-Streptavidin (Horizon V500, 1:800, BD), anti-CD117(c-kit) (APC/eF780, clone 2B8, 1:1600, eBioscience), anti-Ly-6A/E (Sca-1) (PCP/Cy5.5, clone D7, 1:400, eBioscience), anti-CD150 (PE/Cy7, clone TC15-12F12.2, 1:200, Biolegend), anti-CD48 (BV421, clone HM48-1, 1:400, BD), anti-CD45.1 (APC, clone A20, 1:800, Biolegend) and anti-CD45.2 (BV711, clone 104, 1:100) or with anti-Streptavidin (eF710, 1:200, eBioscience), anti-CD117(c-kit) (APC/eF780, clone 2B8, 1:1600, eBioscience), anti-Ly-6A/E (Sca-1) (PCP/Cy5.5, clone D7, 1:400, eBioscience), anti-CD150 (PE/Cy7, clone TC15-12F12.2, 1:200, Biolegend), anti-CD48 (BV421, clone HM48-1, 1:400, BD), anti-CD201/EPCR (PCP/eF710, clone ebio1560, 1:200, eBioscience), anti-CD41 (BV605, clone MWReg30, 1:100, Biolegend), anti-CD45.1 (APC, clone A20, 1:800, Biolegend) and anti-CD45.2 (FITC, clone 104, 1:400).

For more details on antibody dilutions, manufacturers, catalog, and clone numbers please refer to Supplementary Table 2. For a detailed overview of gating strategies and cell population abundance refer to Supplementary Fig. 1. Absolute numbers of lin$^-$ BM cells were determined employing a Miltenyi MACSquant flow cytometer.

**Single cell culture**. Single cells were deposited into 96 well U-bottom plate (TPP) using a BD FACS ARIA II cell sorter. Cells were cultivated in Iscove's Modified Dulbecco's Medium (Gibco) supplemented with 10% FCS (FBS superior, Biochrom), 1% Pen/Strep, stem cell factor (rmSCF 20 ng/ml, PeproTech), Thrombopoietin (rmTPO 20 ng/ml, PeproTech), Interleukin-3 (rmIL-3 20 ng/ml, PeproTech), Erythropoietin (rhEPO, 5U/ml, NeoRecormon, Roche). Culture wells were visually inspected and counted by light microscopy (PrimoVert, Zeiss) and assigned to categories on day 5 of culture (see Fig. 4d). Representative wells were stained with Hoechst 33,342 (5 µg/ml) and imaged by fluorescence microscopy (BZ-X710, Keyence) (Supplementary Fig. 5g).

**Single-cell and bulk RNA sequencing**. Single hematopoietic stem and progenitor cells were index-sorted into 384 well plates containing 0.5 µl of nuclease free water with 0.2% Triton-X 100 and 4 U murine RNase Inhibitor (NEB), spun down, and frozen at −80 °C. For bulk RNA transcriptomes, 5000–20,000 CD48$^{hi}$ or CD48$^{-/lo}$ MkPs (lin$^-$ CD117$^+$ Sca-1$^-$ CD150$^+$ CD41$^+$) were sorted and RNA was isolated using the RNeasy Micro Kit (Qiagen). RNA quality was analyzed using a Agilent 2100 Bioanalyzer (Agilent Technologies) and samples were frozen at −80 °C.

After thawing, total RNA was desiccated to 3 µl and denatured for 3 min at 72 °C in the presence of 5 (2.4 for bulk samples) mM dNTP (Invitrogen), 0.5 µM (240 nM for bulk samples) dT-primer (dT-primer: C6-aminolinker-AAGCAG TGGTATCAACGCAGAGTCGACTTTTTTTTTTTTTTTTTTTTTTTTTTTTTT TTVN, where N represents a random base and V any base beside thymidine) and 1 U (4 U for bulk samples) RNase Inhibitor (NEB). The reverse transcription and addition of the template switch oligo is performed at 42 °C for 90 min after filling up to 10 µl with RT buffer mix for a final concentration of 1× superscript II buffer (Invitrogen), 1 M betaine, 5 mM DTT, 6 mM MgCl₂, 1 µM TSO-primer (TSO-primer: AAGCAGTGGTATCAACGCAGAGTACATrGrGrG, where rG stands for ribo-guanosine) 9 U RNase Inhibitor and 90 U Superscript II. The reverse transcriptase is inactivated at 70 °C for 15 min.

The single stranded cDNA is subsequently amplified using Kapa HiFi HotStart Readymix (Roche) at a 1× concentration together with 0.1 µM UP-primer* under following cycling conditions: initial denaturation at 98 °C for 3 min, 23 cycles (18 cycles for bulk RNA samples) [98 °C 20 s, 67 °C 15 s, 72 °C 6 min] and final elongation at 72 °C for 5 min.

The amplified cDNA was purified using 1× volume of hydrophobic Sera-Mag SpeedBeads (GE Healthcare) resuspended in a buffer consisting of 10 mM Tris, 20 mM EDTA, 18.5 % (w/v) PEG 8000, and 2 M sodium chloride solution. The cDNA was eluted in 12 µl nuclease-free water and the concentration of the samples was measured with a Tecan plate reader Infinite 200 pro in 384 well black flat bottom low volume plates (Corning) using AccuBlue Broad range chemistry (Biotium) for single cell RNA samples; for bulk RNA samples, the cDNA quality and concentration was determined with the Fragment Analyzer (Agilent).For library preparation up to 700 pg cDNA (3 ng cDNA for bulk RNA samples) was desiccated and rehydrated in 1 µl Tagmentation mix (1× TruePrep Tagment Buffer L, 0.1 µl (0.5ul for bulk RNA samples) TruePrep Tagment Enzyme V50; from TruePrep DNA Library Prep Kit V2 for Illumina; Vazyme) and tagmented at 55 °C for 5 min.

Subsequently, Illumina indices are added during PCR (72 °C 3 min, 98 °C 30 s, 13 cycles (12 cycles for bulk RNA samples) [98 °C 10 s, 63 °C 20 s, 72 °C 1 min], 72 °C 5 min) with 1× concentrated KAPA Hifi HotStart Ready Mix and 300 nM dual indexing primers. After PCR, libraries of single cell samples are quantified with AccuBlue Broad range chemistry, equimolarly pooled, and purified twice with 1× volume Sera-Mag SpeedBeads. This was followed by Illumina 50 bp paired-end sequencing on a Novaseq6000 aiming at an average sequencing depth of 0.5 mio reads per cell.

For bulk RNA samples, libraries are purified twice with 1× volume Sera-Mag SpeedBeads and quantified with the Fragment Analyzer, followed by paired-end sequencing of 60–83 million fragments on the Illumina NovaSeq 6000 platform in S4 PE mode.

**Single-cell transcriptome analysis**. Raw reads were mapped to the mouse genome (mm10) and splice-site information from Ensembl release 87[60] with gsnap (version 2018–07-04)[61]. Uniquely mapped reads and gene annotations from Ensembl were used as input for featureCounts (version 1.6.2)[62] to create counts per gene and cell. Outlier cells in quality control metrics (library size, number of detected genes, ERCC, and mitochondrial reads) were excluded. One out of three plates exhibited a large number of poor quality cells, thus quality control metrics from the remaining two plates were used for outlier detection. Normalization was performed using the pooling method[63] combined with scaling normalization within each plate to account for different coverages (multiBatchNorm function in the R batchelor package). Batch effects were regressed out of the data using a linear model (removeBatchEffect method in the R limma package). Mutual nearest neighbor correction was also tested, confirming the results from the linear model (Supplementary Fig. 12). Feature selection was performed using variance modeling based on spike-in transcripts and selecting genes with a non-zero biological variance with an FDR threshold of 0.05, then excluding genes contained in the cell cycle entry in Gene Ontology. 10 principal components were retained and used to compute PHATE embedding[44] and transcriptomic distance. Lineage modules were obtained from ref. [3] and scored using the AddModuleScore function of the R package Seurat. A k-nearest-neighbor graph using ten principal components and $k = 15$ was used to compute the PAGA connectivities[47], thus embedded using the Fruchterman–Reingold layout with a threshold of 0.9. Differential gene expression analysis between MkP subsets was performed on log-normalized values using the FindMarkers function (Wilcoxon rank sum test) in the R package scran, blocked by plate of origin. Code is available from Github (https://github.com/hoefer-lab/Thrombopoiesis_RNAseq).

**Bulk transcriptome analysis**. Basic quality control of the resulting sequencing reads was performed using FastQC (https://www.bioinformatics.babraham.ac.uk/projects/fastqc/), and RNA-SeQC (version 1.1.8)[64]. Reads were aligned to the mm10 reference genome using GSNAP (version 2020-12-16)[61]; the Ensembl gene

annotation version 98 was used for splice site detection[60]. Uniquely aligned reads per gene were counted using featureCounts (version 2.0.1)[62] and the respective Ensembl annotation. Normalization of raw read counts based on library size, and testing for differential gene expression between conditions was performed using the DESeq2 R package (version 1.30.1, R version 3.5.1)[65]. 736 genes with an absolute fold-change ≥ 1 and an adjusted $p$-value ≤ 0.05 were considered significantly differentially expressed. Lineage-specific gene sets were obtained from Msigdb[66] using the MsigdbR package and querying the cell type collection (C8) for lineage-related keywords. Gene set enrichment analysis was performed using the cluster-Profiler R package[67]. The analysis can be reproduced by using the code at https://github.com/hoefer-lab/Thrombopoiesis_RNAseq.

**Mathematical modeling**. We use ordinary differentiation equations to describe the dynamics of label propagation, compartment sizes, and H2B-GFP dilution (for details see Supplementary Methods). Computational inference was performed using the trust-region method to search for the local minimum of the negative log-likelihood. To find the global minimum, the initial parameter values were chosen by Latin hypercube sampling (5000 samples for each model). The tasks above were performed using Data2Ddynamic (D2D) framework, a Matlab-based open source package[68]. The source codes and data used in the modeling can be found at https://github.com/hoefer-lab/Thrombopoiesis_Modeling.

**Reporting summary**. Further information on research design is available in the Nature Research Reporting Summary linked to this article.

## Data availability
The scRNA sequencing data generated in this study have been deposited in the NCBI Geo database under accession number GSE159390. The bulk RNA sequencing data generated in this study have been deposited in the NCBI Geo database under accession number GSE183409. Source data are provided with this paper.

## Code availability
The transcriptome analysis and modeling codes are available at Github (https://github.com/hoefer-lab/Thrombopoiesis_RNAseq and https://github.com/hoefer-lab/Thrombopoiesis_Modeling).

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

## Acknowledgements
T.H. was supported by the German Research Council (DFG) through SFB 873 (B11) and SFB 1129 (B11), Marie Skłodowska-Curie Actions H2020-MSCA-ITN-2017 (Quantitative T-Cell Immunology and Immunotherapy, project number: 764698), and DKFZ core funding. A. Gerbaulet was supported by DFG (GE3038/1-1) and Fritz Thyssen Stiftung (Az.10.19.1.013MN). A.R. and A.D. were funded by DFG (RO2133/10-1) in the setting of FOR2577. M.N.F.M. and C.M.M. were supported by the Dresden International Graduate School for Biomedicine and Bioengineering, granted by DFG in the context of the Excellence Initiative. We thank Christa Haase, Madeleine Rickauer and Livia Schulze for technical assistance and all members of Höfer and Gerbaulet groups for discussion.

## Author contributions
A.G. and T.H. conceptualized the study. Experimental work and data analysis was carried out by M.N.F.M. and C.M.M. with assistance from N.D. T.H., C.L., and N.B.B. conceptualized and performed mathematical modeling and statistical inference. Sequencing was performed by S.R. and A.D. A. Greco and K.S. performed transcriptome analyzes. A.R. advised the study. T.H., A.G., M.N.F.M., C.M.M., and C.L. wrote the manuscript. All authors read, commented, and agreed on the final version of the manuscript.

## Funding

## Competing interests
The authors declare no competing interests.
