## [Peer Review File · Nature Communications]

Fate mapping reveals two pathways of native thrombopoiesisThis manuscript has been previously reviewed at another journal that is not operating a transparent peer review scheme. This document only contains reviewer comments and rebuttal letters for versions considered at Nature Communications.

REVIEWER COMMENTS

Reviewer #2 (Remarks to the Author):

The authors have added important new work to supplement their conclusions. The extended transplantation and single-cell functional characterization, and the analyses of fluxes in the aged setting are all very much appreciated, and quite exciting to follow up. At this point, I believe the authors have addressed most of my comments with success.

My only recommendation here is that at some point in the discussion, the authors should make it a bit more explicit that the MkP gates that are being explored contain a relevant (up to 20%) fraction of precursors that appear slightly less committed by scRNAseq analysis and may even show some oligopotent behavior, with traces of CMP-like or My-restricted behavior upon transplantation (Fig 4e), even though their Platelet bias is very clear. I don't think this negatively affects any of the conclusions made from flux analysis in the paper, but I think mentioning this at some point could help better interpret some of the single-cell (expression, culture/transplant) experiments. I would even suggest that maybe we need better markers for definitively committed MkPs (if such a thing even exists)!!

Lastly, I must admit that I had some similar observations (i.e. the CD48^{lo}MkP connection) using the same Fgd5 model a few years ago, but I never managed to put all the pieces of the puzzle together. I'm thrilled that the authors reached the same observations in a completely independent manner and then properly characterized this puzzling and exciting finding, making sense of a truly complicated subject.

Congrats to all involved for a beautiful story.

Reviewer #3 (Remarks to the Author):

The authors have improved a lot their manuscript and have strengthened their major conclusion about the two paths for thrombopoiesis. I have still some major comments about some conclusions of the paper. I strongly recommend the authors to tune done some of their conclusions or remove some conclusions that are deviation from the main message.

1/My major point 1 of previous review about definition of tip-HSC that overlap in part with the major comments 1 and 2 of reviewer 1 has not been fully addressed.

I think the authors should tune done their conclusions "These data identify ES HSC as the tip stem cells of physiological hematopoiesis" page 4. What they need is to established that the labeling propagation strategy allows to infer pathways. Defining whether the ES-HSC is the purest HSC definition and at the top of the hierarchy is not the important points for their study that reveals that the previously identified paths to produce platelets goes through two Mk progenitors and that these two progenitors have different kinetic and mobilization under challenge. Here, the important point is to say that the ES HSC are stably labelled over time and fill the other compartments. Some rephrasing is necessary here.

Another important point about the unspecificity of the labeling is whether it can affect the conclusions about the existence of two type of MkP based on label propagation labeling. I'm not entirely convince by the argument that the labeling of other population than HSC is only on a minor fraction of the cells. This minor fraction of the cells could still be proliferating a lot and/or have a long half-life which can induce some artifact in the label propagation study. I cannot find the information whether the labeling of other population than tip-HSC and HSC are considered into the model. This could be tested at least computationally to see the effect on the conclusions made. Without this type of control, I find the conclusions still debatable.

2/ Another suggestion is to call HSPC what is currently called HSC to overcome debate whether these cells have stem cells properties. Then the conclusion that "myeloid pathway diverges within HSC" would become within HSPC which fits well with what they observe and what is observed in the literature.

3/ Ageing part (page 6) is an interesting side conclusion from their study but is not the focus of the paper. I find that it raises more questions than anything and it is not necessary to the paper. Indeed, the mechanism of myeloid biases with age is a long-standing controversy from the field. The conclusion that the bias comes from an increase production of CMP over CLP would require further validation. I advise to either perform validations of this conclusion or remove it from the paper.

For example, they use ES-HSC as the source in aged mice but no data is validating this hypothesis from the model for old mice. Another example is that the authors didn't comment on the fact that the HSC gain label faster than before. Why conclude only about CMP and CLP? I believe this part is overlooked and need more strengthening if to be included.

Reviewer #4 (Remarks to the Author):

In this manuscript, Morcos and co-authors use a mouse experimental model in combination with mathematical modelling to study the differentiation dynamics of adult hematopoietic stem cells at native physiological settings. The major finding of their study is identification of two distinct pathways of native thrombopoiesis. Two MkP subpopulations were defined through the mathematical modeling, and function studies had confirmed that those two MkP subpopulations can give rise to platelets throw different pathways. Overall, the main conclusions of this study are well-supported by the experimental data and mathematical analysis. The previously reviewers raised a lot of interesting/useful questions/suggestions and comments, which the authors took serious steps including experimental efforts and new data analysis to answer those questions/suggestions and comments, I think the manuscript is clearly improved with more data added and more discussion inferred.

I only have one comment here, in the abstract line 26, the author wrote "Thrombopoiesis is the sum of two pathways that contribute equally in steady state," however I feel there is not enough quantitative data to support these two pathways are "equal" .

Dresden/Heidelberg, 20/01/2022

**Morcos, Li, Munz et al. "Fate mapping reveals two pathways of native thrombopoiesis"
Point-by-point reply to the reviewers' comments (1st revision, 14/12/2021)**

We thank all reviewers for the careful reading of our manuscript and their very supportive and constructive comments. We have addressed all comments raised in a revised version, as detailed in the following.

Reviewer #2 (Remarks to the Author):

The authors have added important new work to supplement their conclusions. The extended transplantation and single-cell functional characterization, and the analyses of fluxes in the aged setting are all very much appreciated, and quite exciting to follow up. At this point, I believe the authors have addressed most of my comments with success.

My only recommendation here is that at some point in the discussion, the authors should make it a bit more explicit that the MkP gates that are being explored contain a relevant (up to 20%) fraction of precursors that appear slightly less committed by scRNAseq analysis and may even show some oligopotent behavior, with traces of CMP-like or My-restricted behavior upon transplantation (Fig 4e), even though their Platelet bias is very clear.

I don't think this negatively affects any of the conclusions made from flux analysis in the paper, but I think mentioning this at some point could help better interpret some of the single-cell (expression, culture/transplant) experiments. I would even suggest that maybe we need better markers for definitively committed MkPs (if such a thing even exists)!!

We thank the reviewer for his very positive and encouraging evaluation of our revised manuscript. His comment on MkP markers is well-taken. Of course, surface markers are generally found to enrich for functional cell populations rather than defining them exclusively. We agree that future efforts focused on defining better markers for committed MkPs would greatly advance the field and hope that our characterization of CD48 as a marker for MkP heterogeneity is a step along this path.

We agree that our phenotypic MkP populations contain cells with residual oligopotency. On page 8, line 251-253, we now state more precisely:

"Functional characterization of both MkPs by culture and transplantation (Fig. 4d-e, and Supplementary Fig. 8f-h) confirmed their predominant thrombocytic potential with some residual oligopotent behavior."

Lastly, I must admit that I had some similar observations (i.e. the CD48^{lo}MkP connection) using the same Fgd5 model a few years ago, but I never managed to put all the pieces of the puzzle together. I'm thrilled that the authors reached the same observations in a completely independent manner and then properly characterized this puzzling and exciting finding, making sense of a truly complicated subject.

Congrats to all involved for a beautiful story.

Reviewer #3 (Remarks to the Author):

The authors have improved a lot their manuscript and have strengthened their major conclusion about the two paths for thrombopoiesis. I have still some major comments about some conclusions of the paper. I strongly recommend the authors to tune down some of their conclusions or remove some conclusions that are deviation from the main message.

1/My major point 1 of previous review about definition of tip-HSC that overlap in part with the major

comments 1 and 2 of reviewer 1 has not been fully addressed.

I think the authors should tune down their conclusions “These data identify ES HSC as the tip stem cells of physiological hematopoiesis” page 4. What they need is to establish that the labeling propagation strategy allows to infer pathways. Defining whether the ES-HSC is the purest HSC definition and at the top of the hierarchy is not the important point for their study that reveals that the previously identified paths to produce platelets goes through two Mk progenitors and that these two progenitors have different kinetic and mobilization under challenge. Here, the important point is to say that the ES HSC are stably labelled over time and fill the other compartments. Some rephrasing is necessary here.

We thank the reviewer for raising this important point and we agree that ES HSCs exhibit stable labeling frequency over time and replenish all other HSC compartments. We re-phrased the sentence on page 4:

“Taken together, these data show that ES HSCs are a self-renewing HSC population that replenish the remainder of HSCs during physiological hematopoiesis, thus placing ES HSCs upstream of other phenotypic HSCs in the hematopoietic hierarchy.”

Another important point about the unspecificity of the labeling is whether it can affect the conclusions about the existence of two types of MkP based on label propagation labeling. I’m not entirely convinced by the argument that the labeling of other population than HSC is only on a minor fraction of the cells. This minor fraction of the cells could still be proliferating a lot and/or have a long half-life which can induce some artifact in the label propagation study. I cannot find the information whether the labeling of other population than tip-HSC and HSC are considered into the model. This could be tested at least computationally to see the effect on the conclusions made. Without this type of control, I find the conclusions still debatable.

We agree with the reviewer that the question of label specificity is very important to our conclusions. As the reviewer acknowledges, we have shown that in populations other than HSCs, only a minor fraction of cells is labeled 10 days after tamoxifen induction (which could either stem from direct, undesired Cre recombination or fast differentiation of labeled HSCs). The reviewer asks whether this minor fraction could, nevertheless, compromise the accurate detection of label propagation from HSCs if the respective populations were rapidly proliferating and/or have a long half-life. For our data, the very small but significant initial labeling of populations other than HSCs, namely MPPs and MkPs (see Supplementary Fig. 2f), does not have a measurable impact on label propagation. To demonstrate this, we followed the reviewer’s suggestion and added initial labeling of MPPs and MkPs (see Supplementary Fig. 2f) to the simulation of the selected model. We found that the effect on label propagation is within experimental error and does not change the dynamics of label propagation.

We added a new Supplementary Figure 9h, showing our original model and fittings from simulated data with initial labelling of MPPs and MkPs side by side. In addition, we now state on page 8:

“Additionally, the small fraction of initially labelled MPPs and MkPs did not have a significant impact on label propagation in the model, supporting that HSCs are the relevant source of fate-mapping label for all assayed lineages (Supplementary Fig. 9h).”

2/ Another suggestion is to call HSPC what is currently called HSC to overcome debate whether these cells have stem cell properties. Then the conclusion that “myeloid pathway diverges within HSC” would become within HSPC which fits well with what they observe and what is observed in the literature.

We agree with this reviewer that functional HSCs are frequently intermingled with phenotypic HSCs. To avoid this confusion we now changed the sub-heading on page 5 to the following:

“The myeloid differentiation pathway diverges within phenotypic HSCs”

3/ Ageing part (page 6) is an interesting side conclusion from their study but is not the focus of the paper. I find that it raises more questions than anything and it is not necessary to the paper. Indeed, the mechanism of myeloid biases with age is a long-standing controversy from the field. The conclusion that the bias comes from an increase production of CMP over CLP would require further validation. I advise to either perform validations of this conclusion or remove it from the paper. For example, they use ES-HSC as the source in aged mice but no data is validating this hypothesis from the model for old mice. Another example is that the authors didn't comment on the fact that the HSC gain label faster than before. Why conclude only about CMP and CLP? I believe this part is overlooked and need more strengthening if to be included.

We respectfully disagree with reviewer #3, we feel that the fate mapping of aged mice significantly strengthens the manuscript. Therefore, we decided not to remove this data set. Regarding the criticism on ES HSCs being the source of all other HSC compartments, we would like to point out that the labeling frequency of ES HSC remained constant over time in our cohort of aged mice. We added two new panels to Supplementary Figure 7 showing the label in ES HSC in old mice over time (Supplementary Figure 7b, c). The apparently faster label acquisition of the total HSC population resulted from the increased fraction of ES HSCs in aged mice (31.1 ± 3.7 in aged mice vs. 19.7 ± 2.8 in young mice, now shown in Supplementary Figure 7c). The well-described expansion of the HSC compartment is primarily driven by an increase in ES HSC numbers. Therefore, the absolute labeling frequency of total HSCs was significantly higher in aged *Fgd5^{ZsGreen:CreERT2}/R26^{LSL-tdRFP}* mice. This increased proportion of ES HSCs as well as the concomitant decrease of non-ES HSC fueled faster label equilibration within the total HSC population. However, propagation of label to HSPC populations downstream of HSCs (e.g. MPPs) remained highly similar in young and ageing mice. The differentiation of HSCs into MPPs remained the rate limiting step of label propagation in young and aged mice.

We added the following explanatory sentences to the manuscript on page 5:

The share of ES HSCs among the total HSC population was significantly larger in aged animals (Supplementary Fig. 7c). Nevertheless, the labelling frequency of ES HSCs in old and young animals remained constant over time (Supplementary Fig. 7b), and the fundamental dynamics of label propagation from HSCs to downstream progenitor compartments were highly similar between aged and young mice (Supplementary Fig. 7d).

Reviewer #4 (Remarks to the Author):

In this manuscript, Morcos and co-authors use a mouse experimental model in combination with mathematical modelling to study the differentiation dynamics of adult hematopoietic stem cells at native physiological settings. The major finding of their study is identification of two distinct pathways of native thrombopoiesis. Two MkP subpopulations were defined through the mathematical modeling, and function studies had confirmed that those two MkP subpopulations can give rise to platelets through different pathways. Overall, the main conclusions of this study are well-supported by the experimental data and mathematical analysis. The previously reviewers raised a lot of interesting/useful questions/suggestions and comments, which the authors took serious steps including experimental efforts and new data analysis to answer those questions/suggestions and comments, I think the manuscript is clearly improved with more data added and more discussion inferred.

I only have one comment here, in the abstract line 26, the author wrote “Thrombopoiesis is the sum

of two pathways that contribute equally in steady state,” however I feel there is not enough quantitative data to support these two pathways are “equal” .

We thank the reviewer for his/her positive evaluation of our work.

We agree that the contribution of both pathways to steady state thrombopoiesis may not be completely equal (Fig. 4h). We changed the abstract to the following:

“Thrombopoiesis is the sum of two pathways that make comparable contributions in steady state.”

REVIEWER COMMENTS

Reviewer #3 (Remarks to the Author):

I am happy with the changes made by the authors except for the ageing part. I still believe this part should be removed from the paper as it is a detour from the main message. One illustration of this is that all the data are only included in a supplementary figure. If the authors insist in including this part then here are my detailed comments about it.

First the results from this part are not validated experimentally and rely only of the findings from model. This particularly important as no quality control of the model fitting to the data is provided, precluding us to judge if it is as good as in the other part of the paper. Also the statement « the fundamental dynamics of label propagation from HSCs to downstream progenitor compartments were highly similar between aged and young mice (Supplementary Fig. 7d). » is a very qualitative statement. I do see little variation that could have consequences for the result from the model.

Basic data are missing such as an illustration of an increase in myeloid cells in this mouse model with age as well as an increase in phenotypic HSC frequency and numbers.

I'm happy with the new addition of data about the frequency of labeling of ES-HSC in aged mice. However to conclude that the increase in HSC is "primarily driven by ES-HSC" the total number and not only % should be displayed.

On the rates from the model that changed with aged, the authors conclude "found that the differentiation rates from CD201^{lo}/Sca-1^{hi} HSCs to Sca-1^{lo} 191 MPPs were substantially increased", it is true but there is also an increase of CD201^{high} sca1^{high} HSC \diamond CD201^{low} sca1^{high} HSC and CD201^{high} sca1^{high} HSC \diamond CD201^{low} sca1^{low} that is not commented in the text. Why is this not commented? Similarly, the increase in the rate of CD48^{low} MkP \diamond Mk without change in sca^{low} CD201^{low} HSC \diamond CD48^{low} MKP is not commented if significant. The author gives therefore a partial and biased conclusion on the data.

I'm maybe wrong but it is strange that all the paths that are different are all increased in aged compare to young mice and no decrease is observed. I would assume that if something goes up another will go down.

The H2B GFP data are displayed but never commented. What are they used for here?

**Morcos, Li, Munz et al. "Fate mapping reveals two pathways of native thrombopoiesis"
Point-by-point reply to the reviewer's comments (2nd revision, 23/02/2022)**

We thank Reviewer #3 for the careful reading of our manuscript and his/her supportive and constructive comments. We have addressed all remaining comments raised in a revised version, as detailed in the following.

Reviewer #3

1. *I am happy with the changes made by the authors except for the ageing part. I still believe this part should be removed from the paper as it is a detour from the main message. One illustration of this is that all the data are only included in a supplementary figure*

The reviewer remains concerned that the fate mapping and mitotic tracking data obtained in aged mice detract from the main message of the manuscript, namely the description of two distinct pathways of thrombopoiesis. We appreciate the concern to not overload our manuscript. However, we would like to emphasize that these data add substantially to the main message. Our finding that aged mice increase the use of our newly described, direct pathway to platelets via CD48^{-/lo} megakaryocyte progenitors pertains directly to the core message of the paper. Hence this result is shown in a main figure, Figure 4i, as is the related result on enhanced myeloid bias (Figure 2i). Hence, key functional differences we identified in hematopoietic differentiation between young and old mice are shown in main figures. That we show the underlying primary data on aged mice in Supplementary Fig. 7 is simply to avoid repetition, as the methodology applied to measure and analyze these data is exactly the same as we have used for young mice and which is documented in the main figures. In support of our view, we would like to add that Reviewer #2 has stated in his final report: "The extended transplantation and single-cell functional characterization, *and the analyses of fluxes in the aged setting* are all very much appreciated".

2. *First the results from this part are not validated experimentally and rely only of the findings from model. This particularly important as no quality control of the model fitting to the data is provided, precluding us to judge if it is as good as in the other part of the paper.*

The reviewer asks about the quality of the model fit to the data. We agree that is a valid point and have now included the fit of the model to the fate mapping and mitotic tracking data in aged mice (Supplementary Figure 7j), which is as good as the fit in young mice (shown in Figure 2). Of note, we applied the same rigor to the analysis of fate mapping and mitotic tracking data obtained in aged mice as we did to the analysis of these data from young mice. We stress that both data sets are of the same high quality (Supplementary Figures 7f-i show the primary data side by side). This can be appreciated by the overall narrow confidence bounds of the cell fluxes computed from both sets of data, implying that the combined data contain information on these fundamental parameters both in young and aged mice (Supplementary Fig. 7k).

3. *Also the statement « the fundamental dynamics of label propagation from HSCs to downstream progenitor compartments were highly similar between aged and young*

mice (Supplementary Fig. 7d). “ is a very qualitative statement. I do see little variation that could have consequences for the result from the model.

The reviewer criticizes the merely qualitative nature of the above statement. To make the text more precise, we have made the following additions:

First, we have complemented the above sentence with a statement that despite similar label propagation, the labeling increased faster “in total HSCs, MkPs, and platelets in aged animals”, (page 6, line 187) which is seen by quantifying both sets of data side-by-side in Supplementary Figure 7f-l. Second, to provide the desired quantitative statements, we now point out all differentiation rates that are significantly different in aged compared to young mice at fitting positions in the manuscript as follows:

(1) from ES HSCs to total HSCs, from CD201^{-/lo} Sca-1^{hi} HSCs to Sca-1^{lo} MPPs and from Sca1^{lo} HPC-1 to CMPs on page 6, line 192:

We found that the differentiation rates from CD201^{-/lo} Sca-1^{hi} HSCs to Sca-1^{lo} MPPs and from Sca1^{lo} HPC-1 to CMPs were significantly increased (Supplementary Table 1), as did the respective cell fluxes (Supplementary Fig. 7k); by contrast, the faster label propagation from ES HSCs to the remainder of the HSCs (Supplementary Fig. 7k) was accounted for by the increase in ES HSC number (Supplementary Fig. 7l).

and (2) from CD48^{-/lo} megakaryocyte progenitors to megakaryocytes on page 9, line 288:

... revealed an increased contribution of the CD48^{-/lo} MkP pathway, accounting for about 80% of thrombopoiesis (Fig. 4i).

4. *Basic data are missing such as an illustration of an increase in myeloid cells in this mouse model with age as well as an increase in phenotypic HSC frequency and numbers. I'm happy with the new addition of data about the frequency of labeling of ES-HSC in aged mice. However to conclude that the increase in HSC is “primarily driven by ES-HSC” the total number and not only % should be displayed.*

The reviewer asks about experimental data probing myeloid bias in aged mice and increase in phenotypic HSCs. To the first point, we stress that myeloid repopulation bias has been reported upon transplantation of aged HSC (Sudo et al, J Exp Med 2000, Müller-Siegburg et al. Blood 2004, Dykstra et al. J Exp Med 2011). However, we would like to add respectfully that the reviewer is not correct in expecting an increase in myeloid cells in peripheral blood of 18-month old non-transplanted mice. Consistent with the literature, we find that this is not the case (please see reviewer figure R1). What we find is an increase in myeloid output of HSCs. This has previously been described for transplantation, and our data now show that this is also the case for native hematopoiesis. Increased myeloid HSC output at constant myeloid cells numbers in the periphery would be consistent with shorter lifetime or altered tissue distribution of the latter in aged mice.

In response to the second point, we indeed see an increase of immuno-phenotypic HSCs in the bone marrow (Supplementary Fig. 7b) as well as increased platelet numbers in peripheral blood (Supplementary Fig. 7c). These data substantiate the expected aging phenotype of hematopoiesis in our aged animal cohort. We mention this on page 6, line 181:

As previously described, aged mice exhibited increased HSC frequency^{35,36} and thrombocytopenia³⁷ (Supplementary Fig. 7b-c).

and again on page 9, line 289:

Consistently, we observed increased platelet numbers in aged mice (Supplementary Fig. 7c).

5. *On the rates from the model that changed with aged, the authors conclude “found that the differentiation rates from CD201^{-/lo} Sca-1^{hi} HSCs to Sca-1^{lo} MPPs were substantially increased”, it is true but there is also an increase of CD201^{high} sca1^{high} HSC CD201^{low} sca1^{high} HSC and CD201^{high} sca1^{high} HSC CD201^{low} sca1^{low} that is not commented in the text. Why is this not commented? Similarly, the increase in the rate of CD48^{low} MkP Mk without change in sca^{low} CD201^{low} HSC CD48^{low} MKP is not commented if significant. The author gives therefore a partial and biased conclusion on the data.*

Reviewer #3 observed correctly that additional differentiation fluxes are altered in aged mice. This information has been available in the manuscript (graphically in Supplementary Figure 7k, and in numerical detail in Supplementary Table 1). We agree that for clarity all significant flux changes deserve to be mentioned in the text (in the statement that the reviewer cites, we highlighted the particular rate that accounts for the increased myeloid bias). Hence, we added a comprehensive statement on these altered fluxes, as detailed in our response to Point 3.

6. *I’m maybe wrong but it is strange that all the paths that are different are all increased in aged compare to young mice and no decrease is observed. I would assume that if something goes up another will go down.*

We understand that the reviewer suggests that as the fluxes of some differentiation processes go up with aging, the fluxes of other processes should go down. It turns out that this assumption does not have a mechanistic basis. For example, the increased cell fluxes from HSCs reflect the rise in HSC number in aged mice. The increased flux from CD48^{-/lo} megakaryocyte progenitors to megakaryocytes is due to a joint increase in differentiation and self-renewal of the former. The idea of decreases compensating for increases does not apply to stem cell differentiation.

7. *The H2B GFP data are displayed but never commented. What are they used for here?*

As explained in detail (Section entitled “Combining HSC fate mapping with mitotic tracking yields lineage pathways”, Supplementary Figure 4, Supplementary Methods), the H2B-GFP proliferation data was used in combination with the label propagation data to fit the model. To stress this point, we now added on page 6, line 190:

Using the mathematical model (Fig. 2c), we inferred the differentiation, proliferation and death rates of aged HSPCs from the combined fate mapping, mitotic tracking and cell number data, based on an accurate model fit within experimental error (Supplementary Fig. 7j).

Figure R1: Changes in frequency of myeloid peripheral blood cells in aged and young mice

Young (n=5, 16 weeks) and old (n=10, 52-64 weeks) *Fgd5^{ZsGreen:CreERT2}/R26^{LSL-tdRFP}* mice were TAM- induced and analyzed 17 weeks later; Percentage of CD11b+ cells and neutrophils (Gr-1+ cells among CD11b+ cells) among viable (PI-) peripheral blood cells are shown.

REVIEWERS' COMMENTS

Reviewer #3 (Remarks to the Author):

The authors have improved significantly the again part of the paper and have answered all my comments.